# Deep graph kernel point processes

## Abstract

Point process models are widely used for continuous asynchronous event data, where each data point includes time and additional information called "marks", which can be locations, nodes, or event types. In this paper, we present a novel point process model for discrete event data over graphs, where the event interaction occurs within a latent graph structure. Our model builds upon the classic influence kernel-based formulation by Hawkes in the original self-exciting point processes work to capture the influence of historical events on future events' occurrence. The key idea is to represent the influence kernel by Graph Neural Networks (GNN) to capture the underlying graph structure while harvesting the strong representation power of GNN. Compared with prior works that focus on directly modeling the conditional intensity function using neural networks, our kernel presentation herds the repeated event influence patterns more effectively by combining statistical and deep models, achieving better model estimation/learning efficiency and superior predictive performance. Our work significantly extends the existing deep spatio-temporal kernel for point process data, which is inapplicable to our setting due to the fundamental difference in the nature of the observation space being Euclidean rather than a graph. We present comprehensive experiments on synthetic and real-world data to show the superior performance of the proposed approach against the state-of-the-art in predicting future events and uncovering the relational structure among data.

## 1 Introduction

Asynchronous discrete event data, where each data point includes time and additional information called "marks", are ubiquitous in modern applications such as crime (Zhu & Xie, 2022), health care (Wei et al., 2023), earthquake events (Ogata, 1988; Zhu et al., 2021a), and so on. In contemporary applications, the collection of discrete events often reveals an underlying latent graph structure, leading to the widespread adoption of models incorporating graph structures for various purposes. Point processes over latent graphs are a popular model for such data, where the graph nodes can be introduced to capture event marks, which can for example be locations or event types.

Classic one- and multi-dimensional temporal, self-exciting point process models introduced by Hawkes (Hawkes, 1971) leverage an event influence kernel function to capture the impact of historical events on future events' occurrence. The influence kernel takes an exponentially decaying form, often not expressive enough to capture complex influence mechanisms. Recently, there have been many successes in deep point process models that represent the influence kernel using neural networks for temporal-only kernels (Zhu et al., 2021b), spatio-temporal kernels (Okawa et al., 2021), and non-stationary kernels (Dong et al., 2022). Such works achieve competitive performance through efficient modeling of the influence kernel compared to point processes that model the conditional intensity function using neural networks. The key lies in that modeling event influence through a kernel captures the repeated influence patterns more effectively by combining statistical and deep models, thus achieving better model estimation/learning efficiency and superior predictive performance.

Despite much success in prior work on deep kernel modeling of point processes, there has been limited work in exploiting underlying graph structures of multi-dimensional point processes by harvesting the representation power of Graph Neural Networks (GNNs). Although GNNs provide flexible frameworks for modeling graph data, how to properly adopt GNNs into point processes while preserving the statistical model interpretability remains an open question.

In this paper, we present a novel point process model for discrete event data over graphs, where the event interaction occurs within a latent graph structure. We represent the influence kernel by a GNN to

capture the underlying dynamics of event influence. Specifically, the proposed graph-based influence kernel approach provides a unified framework for integrating various GNN structures with point processes via localized graph filter basis functions. It is completely flexible to capture non-stationary inter-node event promotion, inhibition, and multi-hop effects. We also present a computationally efficient and flexible learning scheme by considering two types of loss functions, the commonly used maximum likelihood estimation (MLE) and a new least-square estimation (LSE) scheme. We can allow general types of influence (which can be negative), and the non-negative constraint for the conditional intensity function is ensured by a log-barrier penalty in the loss function.

Our work significantly extends the existing deep spatio-temporal kernel for point process data, which cannot apply in our setting since the observation space is fundamentally different: the spatio-temporal point processes are for events that occurred in geophysical Euclidean space, and point processes over graphs represent vicinity using nodes and edges. Our contributions can be summarized as follows:

1. Our proposed method explicitly models the influence kernel in point processes via GNNs instead of typical intensity-based models. This permits greater expressivity of inter-event-category contributions, including non-stationary, multi-hop exciting, and inhibiting effects. Furthermore, the graph kernel can be directly interpreted, yielding clear information about the relational structure in the modeled graph point process.
2. The proposed deep kernel can be efficiently scaled to large graphs by taking advantage of the localized graph filter basis. The basis allows the deep kernel to go beyond simple distance-based influence for graphs representing events in space, providing a model structure for non-spatial graphs such as traffic or social networks. Meanwhile, a larger class of GNN models can be incorporated within our framework, enabling broader versatility in real-world applications.
3. Comprehensive experiments demonstrate that including the latent graph structure in the deep kernel modeling yields benefits over the state-of-the-art in both simulated and real data settings. Our method applies to a wide array of point process data settings, including events generated by infrastructural, climatic, and social phenomena.

## 1.1 RELATED WORKS

There are various deep learning point process models based on *modeling the conditional intensity function* using neural networks (rather than the influence kernel), such as recurrent neural networks (RNNs) (Du et al., 2016; Mei & Eisner, 2017). Due to advances in attention models for sequential data modeling (Vaswani et al., 2017), RNN approaches have been surpassed by self-attention-based approaches (Zuo et al., 2020; Zhang et al., 2020). These RNN and self-attention methods provide expressive models for the conditional intensity; however, they often lack statistical interpretability coming from the original "influence kernel" approach and they do not consider graph structures.

Various approaches have been explored in point process modeling using graph information. Classical multivariate Hawkes processes (Reinhart, 2018) assume a parametric form of the conditional intensity. A recent work (Fang et al., 2023) develops a novel method of the Group Network Hawkes Process that can account for the heterogeneous nodal characteristics using a parametric network model. Our approach differs from parametric point processes by assuming a deep graph-based influence kernel. Many modern approaches (Yang et al., 2021; Zhang et al., 2021; Zuo et al., 2020; Pan et al., 2023) adopt neural networks incorporated with the graph structure. These studies focus on combining non-graph neural network architectures (*e.g.*, fully-connected neural networks, RNNs, etc.) along with certain graph information (*e.g.*, adjacency matrix), rather than directly incorporating GNN structures in the models. For example, A-NHP (Yang et al., 2021) and THP-S (Zuo et al., 2020) adopt attention-based mechanisms with additional constraints on the learned attention weights posted by the graph structure, which differs from the Graph Attention Network (GAT) (Velickovic et al., 2017). These approaches mainly consider the single-hop, adjacency-based influence over the latent graph. Another attempt of the Geometric Hawkes Process (GHP) (Shang & Sun, 2019) integrates graph convolution RNNs with the Hawkes process and achieves enhanced model expressiveness. Compared with our approach, they use GNN to estimate the parameters of the conditional intensity function with a parametric (exponentially decaying) kernel. In our work, incorporating GNN architectures in the influence kernel without any parametric constraints permits the flexible and interpretable recovery of complex event dependency, such as multi-hop influence mechanisms over the graph topology.

Our work is related to GNNs, which have seen wide applications in areas including temporal phenomena on graphs (Longa et al., 2023; Wu et al., 2020b). Graph convolutions have been popular recently, and they incorporate spatial or spectral convolutions based upon the adjacency matrix

or graph Laplacian (Bruna et al., 2013). An early attempt applying spatial convolutions is the Diffusion-convolutional Neural Network (DCNN) (Atwood & Towsley, 2016). Other spatial-based approaches also include attention models (Brody et al., 2021; Velickovic et al., 2017; Dwivedi & Bresson, 2020). Prototypical spectral convolutions are utilized in Chebnet (Defferrard et al., 2016) and graph convolutional networks (GCN) (Kipf & Welling, 2016), with recent extensions including auto-regressive moving average (ARMA) spectral convolutions (Bianchi et al., 2021) and the Simple Spectral Graph Convolution (SSGC) (Zhu & Koniusz, 2021). Modern approaches incorporate both local and global features, such as L3Net (Cheng et al., 2020) and General, Powerful, Scalable (GPS) Graph Transformer (Rampášek et al., 2022).

## 2 BACKGROUND

**Self-exciting point process.** A self-exciting point process (Reinhart, 2018) models the occurrence of time-stamped discrete events that depend on the observed history. Consider a simple temporal point process that only models event times. Let $\mathcal{H} = \{t_1, \ldots, t_n\}$ be an observed event sequence, where $t_i \in [0, T] \subset \mathbb{R}$ is the time of $i$-th event. We denote the history before a given time $t$ as $\mathcal{H}_t = \{t_i | t_i < t\}$. The conditional intensity of events is defined as $\lambda(t) = \lim_{\Delta t \downarrow 0} \mathbb{E}\left[\mathbb{N}([t, t+\Delta t])|\mathcal{H}_t\right]/\Delta t$, where the counting measure $\mathbb{N}$ is defined as the number of events occurring in $[t, t+\Delta t]$. For notational simplicity, we omit the dependency of history $\mathcal{H}_t$ in $\lambda(t)$. The well-known Hawkes process (Hawkes, 1971) models the self-excitation effect from history in an additive manner. The conditional intensity function is defined as

$$\lambda(t) = \mu + \sum_{t' \in \mathcal{H}_t} k(t', t),$$

where $\mu$ is the background intensity, and $k$ is the so-called influence kernel measuring the effects of historical events.

In a *marked point process*, each event is associated with an additional attribute called *mark* denoted by $v \in V$. The mark represents specific characteristics of the event and can be either continuous or categorical, such as event location or event type. Let $\mathcal{H} = \{(t_i, v_i)\}_{i=1}^n$ and $\mathcal{H}_t = \{(t_i, v_i)|t_i < t\}$ be the observed event sequence and history before time $t$, respectively. The conditional intensity with influence kernel $k$ can be written as:

$$\lambda(t, v) = \mu + \sum_{(t', v') \in \mathcal{H}_t} k(t', t, v', v). \tag{1}$$

The influence kernel is crucial when learning the conditional intensity $\lambda(t, v)$ from event sequences. A standard and simplified way (Reinhart, 2018) is to represent the kernel $k(t', t, v', v)$ to be a product of spatial interaction with temporal kernel in the form of $a_{v, v'} f(t - t')$, where the coefficient $a_{v, v'}$ captures the influence of node $v'$ on $v$ through a graph kernel, and $f$ is a stationary temporal kernel. We consider general graph kernel representation using GNNs that go beyond the parametric form in the classic point processes literature, thus enabling better characterizing the event dynamics. We have provided a comprehensive background of the graph kernel in Appendix C.

**Graph convolution.** Graph convolutions in graph neural networks (Wu et al., 2020b) extend the convolution strategy to the graph and address the problem of cyclic mutual dependencies architecturally. Graph convolutions fall into two categories: spectral- and spatial-based models. Spectral graph convolutions introduce graph filters $g_\theta$ based on the full eigen-decomposition of the graph Laplacian. The graph signal $X$ is convoluted by $X *_G g_\theta = U g_\theta U^T X$, where $U$ is the matrix of the eigenvectors of the graph Laplacian ordered by eigenvalues. For instance, in Spectral Convolutional GNNs (Bruna et al., 2013), the graph filter $g_\theta = \Theta_{i,j}$ contains a set of learnable parameters that characterize the relations between node pairs. On the other hand, spatial-based graph convolution is performed by information propagation along edges. The weight matrix in each layer is constructed based on the node's spatial relations (*i.e.*, adjacency matrix). Either the localized filter or the weight matrix plays a pivotal role in capturing the nodal dependencies. Various structures of graph convolutions, both spectral and spatial, can be integrated into our proposed influence kernel to describe a wide spectrum of intricate inter-event-category dependencies.

## 3 POINT PROCESSES ON GRAPHS

### 3.1 PROBLEM DEFINITION

The objective of this study is to construct a point process model for the occurrence of multiple types of events within a latent graph structure. Let $G = (V, E)$ denote the underlying graph, where each

node $v \in V$ represents one event type. An undirected edge connecting nodes $u$ and $v$ indicates the existence of potential interaction between type-$u$ and type-$v$ events. Note that the edges merely suggest the support of possible inter-event-category interactions without dictating the directions.

Consider a set of event sequences $\mathcal{S} = \{\mathcal{H}^1, \mathcal{H}^2, \ldots, \mathcal{H}^{|\mathcal{S}|}\}$, where each $\mathcal{H}^s = \{(t_i^s, v_i^s)\}_{i=1}^{n_s}$ is a collection of events $(t_i^s, v_i^s)$ occurring on node $v_i^s$ at time $t_i^s$. Our proposed graph point process is expected to: (i) jointly predict the times and types of forthcoming events based on the observed historical data and (ii) provide an interpretable understanding of the event generation process by revealing the interdependences among multiple types of events and uncovering the latent graph structure with no prior information. Toward this end, we adopt the statistical formulation of conditional intensity in equation 1 and introduce an influence kernel built on localized graph filters in GNNs, aiming to explicitly characterize the complicated contributing relationship between any binary event pair (*e.g.*, excitation, inhibition, or other dynamic influences).

### 3.2 DEEP TEMPORAL GRAPH KERNEL

Modeling the multi-dimensional influence kernel $k$ for intricate event dependency is crucial yet challenging. To go beyond simple parametric forms of the kernel while maintaining the model efficiency, we represent the multi-dimensional kernel by taking advantage of the kernel singular value decomposition (SVD) (Mercer, 1909; Mollenhauer et al., 2020). Specifically, the influence kernel $k(t', t, v', v)$ in equation 2 is decomposed into basis kernel functions as follows:

$$k(t', t, v', v) = \sum_{d=1}^{D} \sigma_d g_d(t', t - t') h_d(v', v), \tag{2}$$

where $\{g_d, h_d\}_{d=1}^{D}$ are sets of basis kernels in terms of event time and type, respectively. The scalar $\sigma_d$ is the corresponding weight (or "singular value") at each rank $d$. Instead of directly learning the multi-dimensional event dependency, we simplify the task by "separately" modeling specific modes of event dependency over time or graph using different basis kernels. It is worth noting that the weighted combination of basis kernels covers a broad range of non-stationary influence kernels used in point processes, and our kernel $k$ is not decoupled over time and graph space. While functional SVD is usually infinite-dimensional, in practice, we can truncate the decomposition as long as the singular values $\sigma_k$ decay sufficiently fast, only considering a finite rank representation.

The temporal basis kernels are carefully designed to capture the heterogeneous temporal dependencies between past and future events. First, the parametrization of temporal kernels $\{g_d\}_{d=1}^{D}$ using displacements $t - t'$ instead of $t$ provides us a low-rank way to approximate general kernels (Dong et al., 2022). To proceed, we approximate $\{g_d\}_{d=1}^{D}$ using shared basis functions:

$$g_d(t', t - t') = \sum_{l=1}^{L} \beta_{dl} \psi_l(t') \varphi_l(t - t'), \quad \forall d = 1, \ldots, D.$$

Here $\{\psi_l, \varphi_l : [0, T] \to \mathbb{R}\}_{l=1}^{L}$ are two sets of one-dimensional basis functions characterizing the temporal impact of an event occurring at $t'$ and the pattern of that impact spread over $t - t'$. The scalar $\beta_{d,l}$ is the corresponding weight. Each of the basis functions $\{\psi_l, \varphi_l\}_{l=1}^{L}$ are represented by a fully-connected neural network. The universal approximation power of neural networks enables the model to go beyond specific parametric forms of the influence kernel or conditional intensity.

### 3.3 GRAPH KERNEL WITH LOCALIZED GRAPH FILTERS

We develop a novel framework for the graph basis kernels by leveraging the localized graph filters in graph convolution to extract informative inter-event-category patterns from graph-structured data. Specifically, the basis kernels $\{h_d\}_{d=1}^{D}$ are represented as follows:

$$h_d(v', v) = \sum_{r=1}^{R} \gamma_{dr} B_r(v', v), \quad \forall d = 1, \ldots, D,$$

where $\{B_r(v', v) : V \times V \to \mathbb{R}\}_{r=1}^{R}$ are $R$ bases of localized graph filters, and $\gamma_{dr}$ is the corresponding weight for each $B_r$. The bases can be constructed either from a spatial or a spectral approach, corresponding to two categories of commonly seen graph convolutions.

Formally, the temporal graph influence kernel $k$ can be represented as:

$$k(t', t, v', v) = \sum_{r=1}^{R} \sum_{l=1}^{L} \alpha_{rl} \psi_l(t') \varphi_l(t - t') B_r(v', v), \tag{3}$$

where $\alpha_{rl} = \sum_{d=1}^{D} \sum_{r=1}^{R} \sum_{l=1}^{L} \sigma_d \beta_{dl} \gamma_{dr}$. To showcase the model flexibility of our graph basis kernel to incorporate various GNN structures, we implement two examples of *L3Net* (Cheng et al., 2020) and *GAT* (Velickovic et al., 2017) in our numerical experiments (Section 4). L3Net provides a unified framework for both spatial- and spectral-based graph convolutions, and GAT can predict the presence or absence of an edge in a graph. Note that our framework is compatible with general GNN layer types, such as those in *Chebnet* (Defferrard et al., 2016) and *GPS Graph Transformer* (Dwivedi & Bresson, 2020). An example is given the Appendix B. Technical details of the GNN incorporation are presented in Appendix C.

By integrating localized graph filters in GNNs, the benefits of our design for the influence kernel $k$ lie in the following concepts. (i) The kernel enables the adoption of various spectral and spatial filter bases. The combination of $R$ bases allows us to represent complex local and global patterns of inter-node influences with great model expressiveness. (ii) Our framework substantially reduces the number of model parameters to $\mathcal{O}(RC|V|)$ for modeling graph-structured point process data with $|V|$ event types, while classic multivariate point processes and other neural point processes typically require more than $\mathcal{O}(|V|^2)$ parameters. Here $C$ represents the average local patch size (Cheng et al., 2020). In practice, we have $C, R \ll |V|$ when dealing with sparse graphs and considering only up to $o$-hop influence (commonly 2 or 3), which significantly improves the scalability of our model when applied to large graphs. Details of the complexity analysis can be found in Appendix C.

*Choice of the network hyperparameters.* In practice, we can treat $\alpha_{rl}$ as one learnable parameter and the model parameter $D$ is absorbed. We also provide two strategies for determining the kernel rank $L$ and $R$ in Appendix G.

## 3.4 MODEL ESTIMATION

Previous studies of point processes primarily use two types of loss functions for model estimation, including the (negative) log-likelihood function (NLL) (Dong et al., 2023; Reinhart, 2018) and the least square loss function (LS) (Bacry et al., 2020; Cai et al., 2022). We present the incorporation of these two losses in our framework, which is not specific to a particular choice of loss and, in fact, can be quite general. For completeness, we present derivations of two loss functions in Appendix D.

**Negative Log-Likelihood (NLL).** The model parameters can be estimated by minimizing the negative log-likelihood of observing event sequences $\mathcal{S}$ on $[0, T] \times V$ (Reinhart, 2018):

$$\min_\theta \ell_{\text{NLL}}(\theta) := \frac{1}{|\mathcal{S}|} \sum_{s=1}^{|\mathcal{S}|} \left( \sum_{v \in V} \int_0^T \lambda(t, v) dt - \sum_{i=1}^{n_s} \log \lambda\left(t_i^s, v_i^s\right) \right). \tag{4}$$

Note that the model parameter $\theta$ is incorporated into the intensity function $\lambda$. Minimizing the negative log-likelihood is equivalent to the maximum likelihood estimation approach, and the model log-likelihood indicates how well the model fits the occurrences of events.

**Least Square (LS) loss.** Another approach based on least square loss (Hansen et al., 2015) can be adopted to estimate the model parameters. The optimization problem given observed events $\mathcal{S}$ on $[0, T] \times V$ is expressed as

$$\min_\theta \ell_{\text{LS}}(\theta) := \frac{1}{|\mathcal{S}|} \sum_{s=1}^{|\mathcal{S}|} \left( \sum_{v \in V} \int_0^T \lambda^2(t, v) dt - \sum_{i=1}^{n_s} 2\lambda(t_i^s, v_i^s) \right). \tag{5}$$

The least square loss function can be derived from the empirical risk minimization principle (Geer, 2000). Intuitively, we expect that the integral of intensity over infinitesimal time intervals containing event times is approximately one, while during non-event times, it is approximately zero.

Since negative values of the influence kernel are allowed for indicating inhibiting effects from past events, an additional constraint is required for ensuring the non-negativity of the conditional intensity function. To achieve this, we leverage the log-barrier method for optimization in point

processes (Dong et al., 2022), which maintains the model interpretability while being computationally efficient. The final objective function to be minimized is formulated as $\mathcal{L}_1(\theta) := \ell_1(\theta) + \frac{1}{w}p(\theta, b)$ and $\mathcal{L}_2(\theta) := \ell_2(\theta) + \frac{1}{w}p(\theta, b)$ for two loss functions, respectively. Here $p(\theta, b)$ is the log-barrier penalization, where smaller intensity values will incur a greater penalty in comparison to larger values. The scalar $w > 0$ is a weight to control the trade-off between log-likelihood and log-barrier, and $b > 0$ is a lower bound of the intensity value over space to guarantee the feasibility of the logarithm. Both loss functions can be efficiently computed in a numerical way, as illustrated by the computational complexity analysis and computation time comparison in Appendix E.

## 4 EXPERIMENT

In this section, we compare our method using a deep graph kernel, referred to as `GraDK`, with seven state-of-the-art point process methods on large-scale synthetic and real-world data sets.

**Baselines.** We include two groups of baselines with distinctive model characteristics. Models in the first group treat the associated node information of each event as one-dimensional event marks without considering the graph structure, including (i) Recurrent Marked Temporal Point Process (`RMTPP`) (Du et al., 2016) that uses a recurrent neural network to encode dependence through time; (ii) Fully Neural Network model (`FullyNN`) (Omi et al., 2019) that models the cumulative distribution via a neural network; and (iii) Deep Non-Stationary Kernel (Dong et al., 2022) with a low-rank neural marked temporal kernel (`DNSK-mtpp`). The second group includes two models that encode the latent graph structure information when modeling event times and marks, including (iv) Structured Transformer Hawkes Process (`THP-S`) (Zuo et al., 2020) and (v) Graph Self-Attentive Hawkes Process (`SAHP-G`) (Zhang et al., 2020) with a given graph structure, which both use self-attention mechanisms to represent the conditional intensity. The comparison with the above baselines enables us to comprehensively investigate the benefits of incorporating latent graph structures in discrete event modeling within a graph, as well as showcase our model capability to capture the complex multi-hop and non-Euclidean event dependencies.

**Experimental setup.** We demonstrate the adaptability of the proposed method to various advanced GNN architectures and loss functions using three different architectures: (i) `GraDK` with L3Net and NLL (`GraDK+L3net+NLL`), (ii) `GraDK` with L3Net and LS (`GraDK+L3net+LS`), and (iii) `GraDK` with GAT and NLL (`GraDK+GAT+NLL`). Details about the experimental setup and model architectures can be found in Appendix H.

### 4.1 SYNTHETIC DATA

We evaluate the efficacy of our model on large-scale synthetic data sets. We generate four data sets using point processes with the following kernels and latent graph structures: (i) a non-stationary temporal kernel on a 3-node graph with negative influence; (ii) a non-stationary temporal kernel on a 16-node graph with 2-hop graph influence; (iii) an exponentially decaying temporal kernel on a 50-node graph with central nodes; and (iv) an exponentially decaying temporal kernel on a 225-node graph. Data sets are simulated using the thinning algorithm (Daley & Vere-Jones, 2007). Each data set contains 1,000 sequences with an average length of 50.9, 105.8, 386.8, and 498.3, respectively. Details regarding synthetic data are presented in Appendix H.

#### 4.1.1 UNOBSERVED GRAPH

We first justify the capability of the proposed framework to recover the structure of the event dependency when we have no prior information about the latent graph structure. This is achieved by incorporating GAT into the graph kernel and uncovering the graph support through the learned attention weights. In particular, we exploit a fully connected graph support in the experiments of `GraDK+GAT+NLL`, assuming possible interactions between any pair of graph nodes. The estimated localized graph filters can indicate the knowledge about the node interactions that the model has learned from the data. As we can see in Figure 1, the recovered graph kernels and event dependencies by `GraDK+GAT+NLL` closely resemble the ground truth. The model's capability to capture the underlying patterns of node interactions or dependencies is crucial in various real-world applications in which one does not have access to the latent graph structure.

#### 4.1.2 OBSERVED GRAPH

We demonstrate the exceptional performance of `GraDK` in modeling point process data by leveraging observed latent graph structures using synthetic data sets. These experiments aim to simulate scenarios where prior knowledge of the latent space is available.

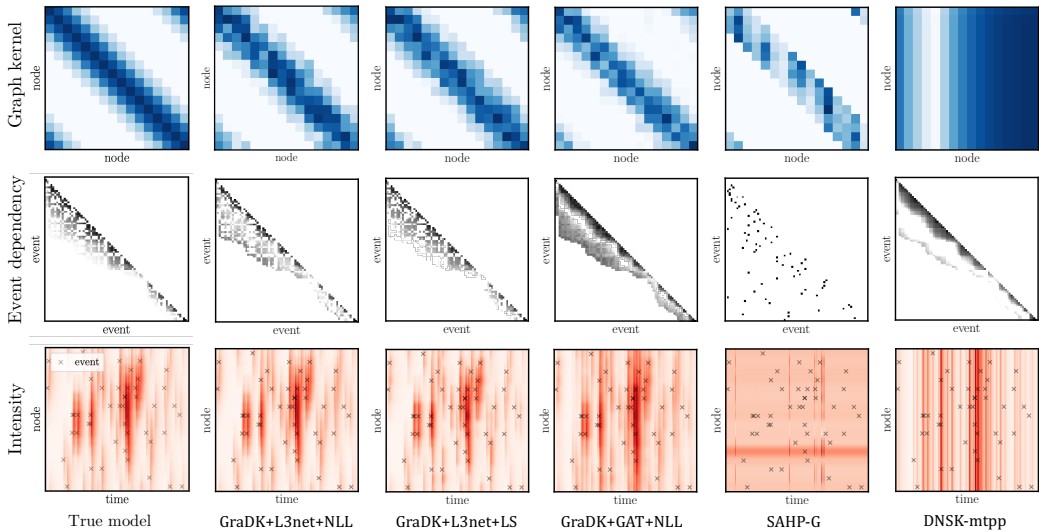

Figure 1: Graph kernel, inter-event dependence, and conditional intensity recovery for the 16-node synthetic data set with 2-hop graph influence. The first column reflects the ground truth, while the subsequent columns reflect the results obtained by `GraDK`, `SAHP-G`, and `DNSK`, respectively.

Table 1: Synthetic data results.

| | 3-node graph with negative influence | | | 16-node graph with 2-hop influence | | | 50-node graph | | | 225-node graph | | |
|---|---|---|---|---|---|---|---|---|---|---|---|---|
| Model | Testing $\ell$ | Time MAE | Type KLD | Testing $\ell$ | Time MAE | Type KLD | Testing $\ell$ | Time MAE | Type KLD | Testing $\ell$ | Time MAE | Type KLD |
| RMTPP | $-3.473_{(0.087)}$ | 0.528 | 0.093 | $-7.239_{(0.193)}$ | 0.301 | 0.142 | $-27.915_{(1.251)}$ | 37.666 | 0.103 | $-16.294_{(1.047)}$ | 29.329 | 0.252 |
| FullyNN | $-2.086_{(0.009)}$ | 0.291 | 0.006 | $-3.347_{(0.018)}$ | 0.198 | 0.018 | $-1.736_{(0.019)}$ | 13.295 | 0.058 | $-3.241_{(0.026)}$ | 9.672 | 0.174 |
| DNSK-mtpp | $-2.127_{(0.003)}$ | 0.149 | 0.012 | $-3.005_{(0.002)}$ | 0.085 | 0.002 | $-1.165_{(0.003)}$ | 1.074 | 0.076 | $-2.511_{(0.001)}$ | 3.958 | 0.086 |
| THP-S | $-2.089_{(0.008)}$ | 0.413 | 0.006 | $-3.079_{(0.004)}$ | 0.108 | 0.011 | $-1.091_{(0.005)}$ | 3.940 | 0.019 | $-2.550_{(0.008)}$ | 4.109 | 0.087 |
| SAHP-G | $-2.113_{(0.005)}$ | 0.172 | 0.003 | $-3.036_{(0.008)}$ | 0.155 | 0.005 | $-1.099_{(0.004)}$ | 1.119 | 0.014 | $-2.506_{(0.005)}$ | 3.578 | 0.094 |
| GraDK+L3net+NLL | $-2.062_{(0.003)}$ | 0.048 | **<0.001** | $-2.995_{(0.004)}$ | 0.028 | 0.002 | $-1.065_{(0.005)}$ | 1.023 | **0.004** | $-2.487_{(0.005)}$ | 1.779 | 0.015 |
| GraDK+L3net+LS | **$-2.059_{(0.002)}$** | **0.021** | **<0.001** | **$-2.993_{(0.003)}$** | **0.001** | **0.001** | **$-1.056_{(0.003)}$** | **0.957** | 0.005 | **$-2.485_{(0.003)}$** | **0.146** | **0.012** |
| GraDK+GAT+NLL | $-2.073_{(0.004)}$ | 0.068 | **<0.001** | $-2.997_{(0.005)}$ | 0.148 | **<0.001** | $-1.690_{(0.001)}$ | 0.690 | 0.007 | $-2.493_{(0.006)}$ | 2.458 | 0.092 |

*Numbers in parentheses are standard errors for three independent runs.

**Kernel and intensity recovery.** Figure 1 contains the recovered graph kernel by each method for the synthetic data generated by the kernel on a 16-node ring graph with 2-hop influence. Our method and `DNSK` directly learn the kernel, and the graph kernel in `SAHP-G` is constructed as in the original paper (Zhang et al., 2020) by computing the empirical mean of the learned attention weights between nodes. It is worth noting that our model learns an accurate representation, reconstructing the self-exciting and multi-hop influential structures in the ring graph, while `SAHP-G` only recovers the mutual dependencies within one-hop neighbors, restricted by their model formulation. The multi-hop influence along the graph structure is also reflected in the true and recovered event intensity by `GraDK` (the first four panels at the bottom row of Figure 1). The conditional intensities of `SAHP-G` and `DNSK`, however, either fail to capture the magnitude of this interdependence or do not accurately decay node dependence along the ring-structure connections.

**Event dependency.** Our model also exhibits exceptional performance in capturing sequential event dependencies. The second row of Figure H6 visualizes the learned inter-event dependency given a sample sequence from the testing set. The dependency between a prior and a future event is characterized by the influence kernel (equation 2) in `GraDK`, `DNSK`, and the true model. For `SAHP-G`, the event dependency is indicated by the scaled self-attention weight (Equation 10 (Zhang et al., 2020)). While `SAHP-G` is capable of discovering long-term dependencies, the decaying influence of recent events is not represented. The event dependency of `DNSK` does well to capture the decaying influence of recent events, but fails to capture long-term effects by certain event types. Our method learns both of these features, capturing long-term dependence and decaying influence similar to that of the true model. Similarly, the second row of Figure 1 shows the inter-event dependency for the data on the 16-node ring graph with 2-hop influence. Still, `SAHP-G` erroneously presents some long-term effects and `DNSK` fails to capture intermediate-time influence from past events, whereas `GraDK` captures the influence at all proper timescales.

**Predictive ability.** The superior predictive performance of `GraDK` is further demonstrated through a comprehensive evaluation. Apart from assessing the fitted log-likelihood ($\ell$) of the testing data, for

Table 2: Ablation study: Comparison with Spatio-Temporal Point Processes (STPP).

| Model | #parameters | Wildfire (25 nodes) | | | Theft (52 nodes) | | |
|---|---|---|---|---|---|---|---|
| | | Testing $\ell$ | Time MAE | Type KLD | Testing $\ell$ | Time MAE | Type KLD |
| MHP | 625 | $-3.846_{(0.003)}$ | 1.103 | 0.072 | $-3.229_{(0.005)}$ | 1.912 | 0.483 |
| ETAS | 2 | $-3.702_{(0.002)}$ | 1.134 | 0.385 | $-3.049_{(0.001)}$ | 3.750 | 0.685 |
| DNSK-stpp | 4742 | $\mathbf{-3.647}_{(0.005)}$ | 0.861 | 0.214 | $-3.004_{(0.002)}$ | 1.342 | 0.600 |
| GraDK | 411 | $-3.650_{(0.002)}$ | **0.580** | **0.018** | $\mathbf{-2.998}_{(0.002)}$ | **0.127** | **0.292** |

*Numbers in parentheses are standard errors for three independent runs.

each data set, we generate 100 event sequences using each learned model (one of three independent runs) and provide two metrics: (i) the mean absolute error of predicted event frequency (*Time MAE*) compared to that in the testing data, and (ii) the Kullback–Leibler Divergence of predicted event types (*Type KLD*), which compares the empirical distributions of event types (nodes) in the testing data and generated sequences. These metrics (proposed in a previous study (Juditsky et al., 2020)) reflect the model's predictive capacity for future events, as opposed to individual event prediction accuracy, which tends to be noisy when applied to large graphs. The quantitative results in Table 1 demonstrate that the `GraDK` method excels in fitting sequential data on a latent graph. It achieves the highest log-likelihood across all datasets and significantly outperforms all baseline methods in predicting future events, which holds immense importance within the domain of point process modeling.

**Comparison of NLL and LS.**    The empirical results indicate that both loss functions can be well-suited for learning graph point processes. In particular, models with LS show consistently better performance by a slight margin. In terms of the model complexity, we show in Appendix E that both loss functions enjoy efficient computation of complexity $\mathcal{O}(n)$, where $n$ is the total number of events, and their computation times are similar.

## 4.2   REAL DATA

We test the performance of our proposed approach on real-world point process data. Since the applications of graph point processes involve discrete events over networks with asynchronous time information, most of the traditional benchmark graph data sets are not applicable. In the following, we collect three publicly available data sets for numerical experiments: (i) traffic congestion data in Atlanta; (ii) wildfire data in California; and (iii) theft data in Valencia, Spain. Details of the real data sets can be found in Appendix H.

### 4.2.1   ABLATION STUDY

For real-world applications involving discrete event data observed in geographic space, spatio-temporal point processes (STPPs) with Euclidean-distance-based influence kernel can be used. Nevertheless, through an ablation study where we compare `GraDK` with three STPP baselines, we showcase that our proposed graph point process model is more flexible and powerful in capturing the more complicated event influence across discretized locations (represented by nodes) that are distant but strongly influence each other. Such influences are captured by direct edges between nodes or through multiple hops, which can be harder to capture using spatio-temporal kernels.

We compare `GraDK` against three STPP baselines including (i) Multivariate Hawkes process (`MHP`) (Hawkes, 1971) (ii) Epidemic Type Aftershock Sequence (`ETAS`) model (Ogata, 1988); (iii) Deep Non-Stationary Kernel with a low-rank spatio-temporal kernel (`DNSK-stpp`). Each baseline represents one type of approach for modeling the spatial effect of historical events. `MHP` discretizes the entire space by several geographic units and models the dependency among units using a spatial kernel matrix. On the contrary, `ETAS` and `DNSK-stpp` adopt parametric and neural-network-based spatial kernels and model the event dependencies over continuous Euclidean space, respectively. We estimate each of the four models with a fixed exponentially decaying temporal kernel on wildfire and theft data sets which are originally observed within Euclidean geographic space.

Table 2 presents the quantitative results of three metrics used in synthetic data sets for each model, which demonstrate the superior performance of `GraDK` in fitting the data and predicting future events. It is worth noting that the performance gain of our model does not rely on the increasing of parameters, indicating the model benefits of the proposed graph kernel framework.

### 4.2.2   COMPARISON WITH BASELINES ON REAL-DATA

Results in Table 3 underscore the efficacy of the `GraDK` approach in acquiring knowledge about graph point processes across a diverse array of real-world domains. These settings cover diverse event dependency dynamics, as the influence mechanisms include infrastructure (roadways for

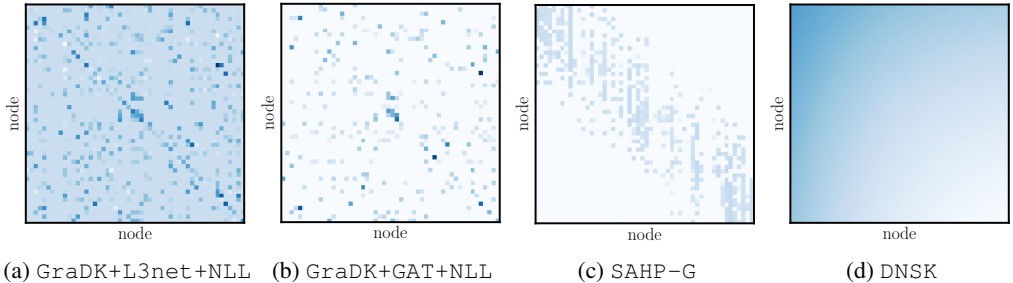

(a) `GraDK+L3net+NLL`  (b) `GraDK+GAT+NLL`  (c) `SAHP-G`  (d) `DNSK`

Figure 2: Learned graph kernels for the theft data set; our proposed method can capture complex inter-node dependence compared with prior work `DNSK` using a spatio-temporal kernel.

Table 3: Real data results.

| Model | Traffic congestion (5 nodes) | | | Wildfire (25 nodes) | | | Theft (52 nodes) | | |
|---|---|---|---|---|---|---|---|---|---|
| | Testing $\ell$ | Time MAE | Type KLD | Testing $\ell$ | Time MAE | Type KLD | Testing $\ell$ | Time MAE | Type KLD |
| `RMTPP` | $-5.197_{(0.662)}$ | 2.348 | 0.021 | $-6.155_{(1.589)}$ | 1.180 | 0.178 | $-11.496_{(1.474)}$ | 5.871 | 0.124 |
| `FullyNN` | $-3.292_{(0.108)}$ | 0.511 | 0.012 | $-4.717_{(0.119)}$ | 0.817 | 0.026 | $-3.468_{(0.068)}$ | 6.457 | 1.169 |
| `DNSK-mtpp` | $-2.401_{(0.011)}$ | 0.934 | 0.010 | $-3.706_{(0.008)}$ | 0.711 | 0.083 | $-3.347_{(0.012)}$ | 0.507 | 0.177 |
| `THP-S` | $-2.254_{(0.007)}$ | 0.378 | 0.003 | $-4.523_{(0.018)}$ | 1.183 | 0.134 | $-2.982_{(<0.001)}$ | 0.739 | 0.189 |
| `SAHP-G` | $-2.453_{(0.013)}$ | 0.729 | 0.021 | $-3.919_{(0.040)}$ | 0.551 | 0.032 | $\mathbf{-2.970}_{(0.032)}$ | 0.464 | 0.096 |
| `GraDK+L3net+NLL` | $-2.178_{(0.005)}$ | 0.314 | **0.001** | $\mathbf{-3.625}_{(0.002)}$ | **0.207** | **0.006** | $-2.980_{(0.003)}$ | 0.640 | 0.079 |
| `GraDK+L3net+LS` | $\mathbf{-2.159}_{(0.004)}$ | **0.247** | **0.001** | $-3.628_{(0.002)}$ | 0.347 | 0.013 | $-2.982_{(0.004)}$ | **0.391** | **0.067** |
| `GraDK+GAT+NLL` | $-2.281_{(0.011)}$ | 0.356 | 0.015 | $-3.629_{(0.007)}$ | 0.898 | 0.085 | $-2.995_{(0.006)}$ | 0.942 | 0.173 |

*Numbers in parentheses are standard errors for three independent runs.

traffic patterns), nature (weather and climate for wildfire patterns), and social dynamics (criminal behavior for theft patterns). Despite the inherent complexity of these scenarios, our method excels in providing a robust framework capable of capturing the intricate dependencies and facilitating accurate predictions, demonstrated by the low Time MAE and Type KLD from our method in each setting, which is better than or comparable to the best baselines in each of the three real data sets. Note that we adopt `GraDK+GAT+NLL` to learn the latent graph structure for each real data set. We then evaluate the models by leveraging the recovered graph supports, which are presented in Appendix H.

In Figure 2, the learned graph kernels of (a) `GraDK+L3net+NLL`, (b) `GraDK+GAT+NLL`, (c) `SAHP-G`, and (d) `DNSK` are visualized for the theft data set. The third panel reveals that `SAHP-G` learns a very noisy graph kernel, resulting in a conditional intensity that depends very slightly on inter-event influence, thus learning a homogeneous Poisson process for each node with a relatively high likelihood. The last panel shows that `DNSK` fails to present meaningful or discernible patterns of self-influence or event-type interdependence. Lastly, `GraDK+GAT` captures complex self-influence and inter-node dependencies, and `GraDK+L3net` recovers multi-hop event influence with the aid of the flexible graph kernel, indicating the complex inhomogeneous dynamics among data.

## 5 CONCLUSION

We develop a novel deep kernel for graph point processes using localized graph filters in GNNs. This construction enables efficient learning of intricate and non-stationary event dynamics on a latent graph structure. The modeling of the kernel enhances interpretability, as one can parse the learned kernel to understand event type interdependence. We demonstrate that our approach outperforms existing methods in terms of dependency recovery and event prediction using simulation and extensive real-data experiments. While we showcase four examples of adopting GNN structures via local filters, we provide a flexible framework that can conveniently incorporate alternative GNN architectures.

One potential limitation is that the kernel representation still assumes additive influence over historical events, a characteristic commonly found in kernel-based methods for point processes. A more general kernel can adopt a more complex non-additive influence structure over events. Another model constraint requiring additional verification stems from the distributional shift between training and testing data. In practice, ensuring the non-negativity of the intensity using log-barrier is achievable when the occurrences of new events follow the same probability distribution as the observed ones, aligning with the fundamental principle in machine learning. However, potential distributional shifts may give rise to the occurrence of frequent future events with negative impacts, which could result in a negative intensity. Addressing the issue of distributional shifts in point process data is a topic that will be deferred for future research.

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

## A    COMPARISON WITH EXISTING APPROACHES

To further elaborate on the contributions of our paper, we provide a detailed comparison between our method and existing approaches that study the point processes in the context of graphs, neural networks, and graph neural networks. Table A1 compares our `GraDK` with existing methods in terms of four aspects that play crucial roles in establishing an effective and high-performing model for marked temporal point processes over graphs, which are also the main contributions of our paper that distinguish our proposed method from those in previous studies.

Table A1: Comparison between our method `GraDK` with other point processes with graphs, neural networks, and graph neural networks.

| Model | No parametric constraint | Modeling influence kernel | Joint modeling event time and mark on graph | Using GNN |
|---|---|---|---|---|
| GNHP (Fang et al., 2023) | ✗ | ✓ | ✓ | ✗ |
| GINPP (Pan et al., 2023) | ✓ | ✗ | ✗ | ✗ |
| RMTPP (Du et al., 2016) | ✗ | ✗ | ✗ | ✗ |
| NH (Mei & Eisner, 2017) | ✓ | ✗ | ✗ | ✗ |
| FullyNN (Omi et al., 2019) | ✓ | ✗ | ✗ | ✗ |
| SAHP (Zhang et al., 2020) | ✓ | ✗ | ✗ | ✗ |
| DNSK (Dong et al., 2022) | ✓ | ✓ | ✗ | ✗ |
| DAPP (Zhu et al., 2021b) | ✓ | ✓ | ✗ | ✗ |
| LogNormMix (Shchur et al., 2019) | ✓ | ✗ | ✗ | ✗ |
| DeepSTPP (Zhou et al., 2022) | ✗ | ✗ | ✗ | ✗ |
| NSTPP (Chen et al., 2020) | ✓ | ✗ | ✗ | ✗ |
| DSTPP (Yuan et al., 2023) | ✓ | ✗ | ✗ | ✗ |
| THP-S (Zuo et al., 2020) | ✓ | ✗ | ✓ | ✗ |
| SAHP-G (Zhang et al., 2021) | ✓ | ✗ | ✓ | ✗ |
| GHP (Shang & Sun, 2019) | ✗ | ✓ | ✗ | ✓ |
| GNPP (Xia et al., 2022) | ✓ | ✗ | ✗ | ✓ |
| GBTPP (Wu et al., 2020a) | ✗ | ✗ | ✗ | ✓ |
| GraDK (Ours) | ✓ | ✓ | ✓ | ✓ |

Being the first paper that *incorporates GNN in the influence kernel without any parametric constraints*, four aspects of our paper's contributions include (also highlighted in Section 1):

1. We have no parametric model constraints, which becomes increasingly important in modern applications for the model to capture complex event dependencies. Prior works (Du et al., 2016; Fang et al., 2023; Shang & Sun, 2019) typically assume a specific parametric form for the model, although can be data- and computationally efficient, which may restrict the model expressiveness for modern applications with complex and large datasets.

2. We model the influence kernel instead of the conditional intensity, which leverages the repeated influence patterns more effectively by combining statistical and deep models with excellent model interpretability. As shown in Table A1, most previous studies fall into the category of modeling the conditional intensity function instead of the influence kernel. While models are expressive, they ignore the benefits of leveraging the repeated influence patterns, and the model interpretability may be limited.

3. We jointly model event time and mark over a latent graph structure, which most existing approaches fail to achieve. Some approaches (Omi et al., 2019; Shchur et al., 2019) enjoy great model flexibility for modeling event times, but thet only focus on temporal point processes and can hardly be extended to mark space. Another body of research (Du et al., 2016; Pan et al., 2023; Shchur et al., 2019) decouples the time and mark space when modeling event distributions for model simplicity and efficiency, which sometimes limits the model flexibility and leads to sub-optimal solutions. More studies model marked events without any graph structure considered. One example is the family of existing spatio-temporal point process approaches (Chen et al., 2020; Dong et al., 2022; Yuan et al., 2023; Zhou et al., 2022) that are inapplicable in the setting of graph point processes.

4. We adopt GNN in the influence kernel, which faces non-trivial challenges and requires a carefully designed model. As shown in Table A1, there has been limited work in exploiting underlying graph structures by leveraging GNNs. Previous studies (Pan et al., 2023; Zhang et al., 2021; Zuo et al., 2020) combine non-graph neural network architectures (*e.g.*, fully-connected neural networks, RNNs, etc.) along with certain graph information (*e.g.*, adjacency matrix), rather than directly incorporating GNN structures in the models. Another attempt (Shang & Sun, 2019) integrates graph convolutional RNN with the influence kernel. However, they focus on using GNN to estimate the parameters of the conditional intensity function with a parametric (exponentially decaying) kernel. Two previous studies (Wu et al., 2020a; Xia et al., 2022) on modeling point processes resort to message-passing GNNs. However, they do not use influence

kernels and focus on a different problem of graph node interaction prediction rather than modeling the dependencies and occurrences of discrete marked events.

It is worth noting that extending the influence kernel onto the graph structure with GNN architectures incorporated is crucial and non-trivial for modeling point processes over graphs. Because (i) the lack of a straightforward definition of distance between event marks makes the distance-based spatial kernel non-sensible on a graph structure, and (ii) simply replacing the distance-based kernel with some scalar coefficients to capture the interaction between each pair of nodes will significantly limit the model expressiveness, especially in modern applications. In this paper, we design the influence kernel motivated by kernel SVD and Mercer's theorem (Mercer, 1909) and innovatively introduce the concept of localized graph filter basis into the kernel to fully take advantage of the universal representation power of GNNs.

## B  EXAMPLE: GRAPH FILTER BASES IN L3NET

Figure B1 illustrates the mechanism of the graph filter bases in L3Net to capture the event dependencies during the modeling of sequential events on an 8-node graph. The neighborhood orders are highlighted as the superscripts of each graph filter basis.

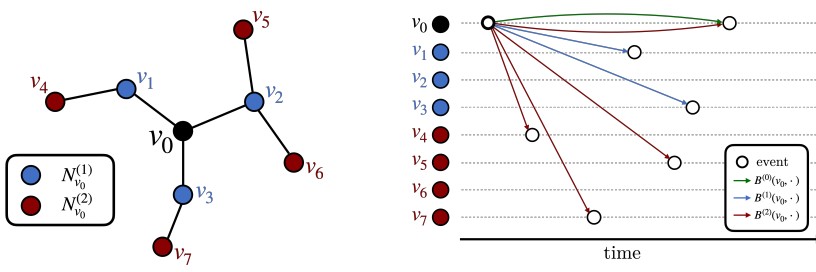

(a) The latent graph structure                (b) Events dependencies over graph nodes

Figure B1: An example of modeling events on an 8-node graph using graph filter bases in L3Net: (a) The latent graph structure. Blue and red nodes represent the 1st and 2nd order neighbors of $v_0$, denoted by $N_{v_0}^{(1)}$ and $N_{v_0}^{(2)}$, respectively. (b) Three graph filter bases $B^{(0)}$, $B^{(1)}$, and $B^{(2)}$ capture the dependencies between events. Hollow circles are events observed on each node. Colored lines indicate the potential influence of the earliest type-$v_0$ event on future events captured by different bases.

## C  INCORPORATION OF LOCALIZED GRAPH FILTERS IN GRAPH KERNEL

We provide the details of incorporating different localized graph filters in graph convolutions into our proposed graph basis kernel, including the model formulation and complexity (number of trainable parameters). A graph convolution layer maps the input node feature $X$ to the output feature $Y$. Particularly, for the graph convolutions in the graph neural networks discussed in Section 3.3, we have:

(1) *Chebnet* (Defferrard et al., 2016): The filtering operation in Chebnet is defined as

$$Y = \sum_{r=0}^{R-1} \theta_r T_r(\tilde{L})X,$$

where $T_r$ is the Chebyshev polynomial of order $r$ evaluated at the scaled and normalized graph Laplacian $\tilde{L} = 2L/\lambda_{\max} - I$, where $L = I - D^{-1/2}AD^{-1/2}$, $D$ is the degree matrix, $A$ is the adjacency matrix, and $\lambda_{\max}$ is the largest eigenvalue of $L$.
We let $B_r = T_{r-1}(\tilde{L}) \in \mathbb{R}^{|V| \times |V|}$, which can be computed by the recurrence relation of Chebyshev polynomial $B_r = 2LB_{r-1} - B_{r-2}$. Since $\{B_r\}_{r=1}^{R}$ are calculated by the graph Laplacian, the trainable parameters are polynomial coefficients (in our case, the weights that combine basis kernels) of $\mathcal{O}(R)$.

(2) *L3Net* (Cheng et al., 2020): In L3Net, the graph convolution layer for single channel graph signal is specified as (omitting the bias, non-linear mapping)

$$Y = \sum_{r=1}^{R} a_r B_r^{(o_r)}(v', v) X.$$

Each $B_r^{(o_r)}(v', v) \neq 0$ only if $v \in N_{v'}^{(o_r)}$.

In our model, we choose an integer $o_r$ for each $B_r$ and formulate them the same way in L3Net. Thus, each $B_r(v', v)$ models the influence from $v'$ to its $o_r$-th order neighbors on the graph. All bases $\{B_r\}_{r=1}^{R}$ are learnable. Assuming the average number of neighbors of graph nodes is $C$, the total number of trainable parameters is of $\mathcal{O}(RC|V|)$.

(3) *GAT* (Velickovic et al., 2017): Considering GAT with $R$ attention heads, the graph convolution operator in one layer can be written as

$$Y = \sum_{r=1}^{R} \mathcal{A}^{(r)} X \Theta_r, \quad \mathcal{A}_{v',v}^{(r)} = \frac{e^{c_{v'v}^{(r)}}}{\sum_{v \in N_{v'}^{(1)}} e^{c_{v'v}^{(r)}}}, \quad c_{v'v}^{(r)} = \sigma((a^{(r)})^{\top}[W^{(r)} X_{v'} || W^{(r)} X_v]).$$

Here $\{a^{(r)}, W^{(r)}\}_{r=1}^{R}$ are trainable model parameters, and $\Theta_r = W^{(r)} C^{(r)}$, where $C^{(r)}$ is a fixed matrix for concatenating outputs from $R$ attention heads. Mask attention is applied to inject the graph structure, that is, $\mathcal{A}_{v',v}^{(r)} \neq 0$ only when $v \in N_{v'}^{(1)}$.

Therefore, each $B_r(v', v)$ in our graph basis kernel can be a learnable localized graph filter with positive entries and column-wise summation normalized to one to mimic the affinity matrix $\mathcal{A}^{(r)}$. When global attention is allowed (*i.e.*, every node can attend on every other node), the number of trainable parameters is of $\mathcal{O}(R|V|^2)$. When mask attention is applied (which is commonly used in GAT), the total number of trainable parameters is $\mathcal{O}(RC|V|)$.

(4) *GPS Graph Transformer* (Rampášek et al., 2022): At each layer of GPS Graph Transformer, the graph node features are updated by aggregating the output of a message-passing graph neural network (MPNN) layer and a global attention layer. Assuming a sum aggregator in the MPNN layer, the filter operation can be expressed as

$$Y = \sum_{r=1}^{R} \left( \mathcal{A}^{(r)} X W_1^{(r)} + X W_2^{(r)} \right) C^{(r)}.$$

Here $\mathcal{A}^{(r)}$ is the affinity matrix in the attention layer, and $W_1^{(r)}, W_2^{(r)}$ are weight matrices in the attention layer and MPNN layer, respectively. The fixed matrices $\{C^{(r)}\}$ concatenate the $R$ attention heads and MPNN layers.

We can integrate such a GPS Graph Transformer structure by introducing $2R$ localized graph filter bases. For each $r$, $B_r$ takes the form of a learnable affinity matrix, and $B_{2r} = A$, where $A$ is the graph adjacency matrix. Here $\{B_r\}_{r=1}^{R}$ are learnable and $\{B_r\}_{r=R+1}^{2R}$ are fixed. The total number of trainable parameters with mask attention adopted is of $\mathcal{O}(RC|V|)$.

## D  DERIVATION OF TWO LOSS FUNCTIONS

Considering one event sequence with $n$ events, we present the detailed derivation of loss functions NLL and LS in the following:

**Negative log-likelihood (NLL).**  The model likelihood of point processes can be derived from the conditional intensity (equation 1). For the $i + 1$-th event at $(t, v)$, by definition, we can re-write $\lambda(t, v)$ as follows:

$$
\begin{aligned}
\lambda(t, v) &= \mathbb{E}\left(\mathbb{N}_v([t, t + \Delta t] \times v)) | \mathcal{H}_t\right) / dt \\
&= \mathbb{P}\{t_{i+1} \in [t, t + \Delta t], v_{i+1} = v | \mathcal{H}_{t_{i+1}} \cup \{t_{i+1} \geq t\}\} / dt \\
&= \frac{\mathbb{P}\{t_{i+1} \in [t, t + \Delta t], v_{i+1} = v, t_{i+1} \geq t, v_{i+1} = v | \mathcal{H}_{t_{i+1}}\} / dt}{\mathbb{P}\{t_{i+1} \geq t | \mathcal{H}_{t_{i+1}}\}} . \\
&= \frac{f(t, v)}{1 - F(t)} .
\end{aligned}
$$

Here $\mathbb{N}_v$ is the counting measure on node $v$, $F(t) = \mathbb{P}(t_{i+1} < t | \mathcal{H}_{t_{i+1}})$ is the conditional cumulative probability and $f(t, v)$ is the corresponding conditional probability density for the next event happening at $(t, v)$ given observed history. We multiply the differential $dt$ on both sides of the equation and sum over all the graph nodes $V$:

$$dt \cdot \sum_{v \in V} \lambda(t, v) = \frac{dt \cdot \sum_{v \in V} f(t, v)}{1 - F(t)} = -d \log \left(1 - F(t)\right).$$

Integrating over $t$ on $[t_i, t)$ with $F(t_i) = 0$ leads to the fact that

$$F(t) = 1 - \exp\left(-\int_{t_i}^{t} \sum_{v \in V} \lambda(t, v) dt\right),$$

then we have

$$f(t, v) = \lambda(t, v) \cdot \exp\left(-\int_{t_i}^{t} \sum_{v \in V} \lambda(t, v) dt\right).$$

Using the chain rule, we have the model log-likelihood to be:

$$\ell_{\text{NLL}} = -\log f((t_1, v_1), \ldots, (t_n, v_n)) = -\log \prod_{i=1}^{n} f(t_i, v_i)$$

$$= \sum_{v \in V} \int_{0}^{T} \lambda(t, v) dt - \sum_{i=1}^{n} \lambda(t_i, v_i).$$

**Least square (LS) loss.** We expect that the integral of the conditional intensity over infinitesimal time intervals containing event times should be approximately one, while during non-event times, it should be approximately zero. Specifically, involving each graph node's counting process $\mathbb{N}_v$, we have

$$\ell_{\text{LS}} = \sum_{v \in V} \int_{0}^{T} (\lambda(t, v) - 1)^2 d\mathbb{N}_v(t) + \int_{0}^{T} (\lambda(t, v) - 0)^2 (dt - d\mathbb{N}_v(t))$$

$$= \sum_{v \in V} \int_{0}^{T} \lambda^2(t, v) d\mathbb{N}_v(t) - 2 \sum_{v \in V} \int_{0}^{T} \lambda(t, v) d\mathbb{N}_v(t) + n + \sum_{v \in V} \int_{0}^{T} \lambda^2(t, v) dt$$

$$- \sum_{v \in V} \int_{0}^{T} \lambda^2(t, v) d\mathbb{N}_v(t)$$

$$= \sum_{v \in V} \int_{0}^{T} \lambda^2(t, v) dt - \sum_{i=1}^{n} 2\lambda(t_i, v_i) + n.$$

Omitting the constant we can have the $\ell_{\text{LS}}$ in equation 5.

The log-likelihood of $|\mathcal{S}|$ observed event sequences can be conveniently obtained by summing over the loss over all the sequences with counting measure $\mathbb{N}_v$ replaced by $\mathbb{N}_s$ for $s$-th sequence.

## E    EFFIECIENT MODEL COMPUTATION

In our optimization problem, the log-barrier term $p(\theta, b)$ guarantees the validity of the learned models by penalizing the value of conditional intensity function on a dense enough grid over space, denoted as $\mathcal{U}_{\text{bar},t} \times V$ where $\mathcal{U}_{\text{bar},t} \subset [0, T]$. The optimization problems with two loss functions are expressed as

$$\min_{\theta} \mathcal{L}_1(\theta) :=$$

$$\frac{1}{|\mathcal{S}|} \sum_{s=1}^{|\mathcal{S}|} \left( \sum_{v \in V} \int_{0}^{T} \lambda(t, v) dt - \sum_{i=1}^{n_s} \log \lambda(t_i^s, v_i^s) \right) - \frac{1}{w|\mathcal{S}||\mathcal{U}_{\text{bar},t} \times V|} \sum_{s=1}^{|\mathcal{S}|} \sum_{t \in \mathcal{U}_{\text{bar},t}} \sum_{v \in V} \log(\lambda(t, v) - b),$$

$$\text{(E1)}$$

and

$$\min_{\theta} \mathcal{L}_2(\theta) :=$$

$$\frac{1}{|\mathcal{S}|} \sum_{s=1}^{|\mathcal{S}|} \left( \sum_{v \in V} \int_{0}^{T} \lambda^2(t, v) dt - \sum_{i=1}^{n_s} 2\lambda(t_i^s, v_i^s) \right) - \frac{1}{w|\mathcal{S}||\mathcal{U}_{\text{bar},t} \times V|} \sum_{s=1}^{|\mathcal{S}|} \sum_{t \in \mathcal{U}_{\text{bar},t}} \sum_{v \in V} \log(\lambda(t, v) - b),$$

$$\text{(E2)}$$

Table E2: Comparison of the computation time for two loss functions on each synthetic data set.

| Model | 3-node graph with negative influence | 16-node graph with 2-hop influence | 50-node graph | 225-node graph |
|---|---|---|---|---|
| GraDK+NLL | 0.931 | 3.993 | 25.449 | 4.948 |
| GraDK+LS | 0.985 | 3.927 | 26.182 | 5.619 |

*Unit: seconds per epoch.

respectively. The computational complexity associated with calculating the objective function has consistently posed a limitation for neural point processes. The evaluations of neural networks are computationally demanding, and traditional numerical computation requires model evaluations between every pair of events. Consider one event sequence $\{(t_i, v_i)\}_{i=1}^n$ in $\mathcal{S}$ with a total number of $n$ events; traditional methods have a complexity of $\mathcal{O}(n^2)$ for neural network evaluation. In the following, we showcase our efficient model computation of complexity $\mathcal{O}(n)$ for two different loss functions using a domain discretization strategy.

**Negative log-likelihood.** We identify three distinct components in the optimization objective (equation E1) as log-summation, integral, and log-barrier, respectively. We introduce a uniform grid $\mathcal{U}_t$ over time horizon $[0, T]$, and the computation for each term can be carried out as follows:

- *Log-summation.* We plug in the influence kernel (equation 3) to the log-summation term and have

$$\sum_{i=1}^n \log \lambda(t_i, v_i) = \sum_{i=1}^n \log \left( \mu + \sum_{t_j < t_i} \sum_{r=1}^R \sum_{l=1}^L \alpha_{lr} \psi_l(t_j) \varphi_l(t_i - t_j) B_r(v_j, v_i) \right).$$

  Each $\psi_l$ is only evaluated at event times $\{t_i\}_{i=1}^n$. To prevent redundant evaluations of the function $\varphi_l$ for every pair of event times $(t_i, t_j)$, we restrict the evaluation of $\varphi_l$ on the grid $\mathcal{U}_t$. By adopting linear interpolation between two neighboring grid points of $t_i - t_j$, we can determine the value of $\varphi_l(t_i - t_j)$. In practice, the influence of past events is limited to a finite range, which can be governed by a hyperparameter $\tau_{\max}$. Consequently, we can let $\varphi_l(\cdot) = 0$ when $t_i - t_j > \tau_{\max}$ without any neural network evaluation. The computation of $B_r(v_j, v_i)$ is accomplished using matrix indexing, a process that exhibits constant computational complexity when compared to the evaluation of neural networks.

- *Integral.* The efficient computation of the integral benefits from the formulation of our conditional intensity function. We have

$$\sum_{v \in V} \int_0^T \lambda(t, v) dt = \mu |V| T + \sum_{i=1}^n \sum_{v \in V} \int_0^T I(t_i < t) k(t_i, t, v_i, v) dt$$

$$= \mu |V| T + \sum_{i=1}^n \sum_{r=1}^R \sum_{v=1}^V B_r(v_i, v) \sum_{l=1}^L \alpha_{rl} \psi_l(t_i) \int_0^{T-t_i} \varphi_l(t) dt.$$

  Similarly, the evaluations of $B_r(v_i, v)$ are the extractions of corresponding entries in the matrices, and $\psi_l$ is only evaluated at event times $\{t_i\}_{i=1}^n$. We leverage the evaluations of $\varphi_l$ on the grid $\mathcal{U}_t$ to facilitate the computation of the integral of $\varphi_l$. Let $F_l(t) := \int_0^t \varphi_l(\tau) d\tau$. The value of $F_l$ on $j$-th grid point in $\mathcal{U}_t$ equals the cumulative sum of $\varphi_l$ from the first grid point up to the $j$-th point, multiplied by the grid width. Then $F_l(T - t_i)$ can be computed by the linear interpolation of values of $F_l$ at two adjacent grid points of $T - t_i$.

- *Log-barrier.* The computation can be carried out similarly to the computation of the log-summation term by replacing $(t_i, v_i)$ with $(t, v)$ where $t \in \mathcal{U}_{\text{bar},t}, v \in V$.

**Least square loss.** Similarly, we term the three components in objective function (equation E2) as square integral, summation, and log-barrier, respectively. The terms of summation and log-barrier are calculated in the same way as the log-summation and log-barrier in equation E1, respectively, except for no logarithm in the summation. Since expanding the integral after squaring the conditional intensity function is complicated, we leverage the evaluations of the intensity function on the dense enough grid for log-barrier penalty $\mathcal{U}_{\text{bar},t} \times V$ and use numerical integration for computing the square integral.

We provide the analysis of the computational complexity of $\mathcal{O}(n)$. For objective equation E1, the evaluation of $\{\psi_l\}_{l=1}^L$ has a complexity of $\mathcal{O}(Ln)$, while the evaluation of $\{\varphi_l\}_{l=1}^L$ requires $\mathcal{O}(L|\mathcal{U}_t|)$

complexity. On the other hand, $\{B_r\}_{r=1}^R$ are computed with $\mathcal{O}(1)$ complexity. Therefore, the total complexity of our model computation is $\mathcal{O}(n + |\mathcal{U}_t|)$. Moreover, the grid selection in practice is flexible, striking a balance between learning accuracy and efficiency. For instance, the number of grid points in $\mathcal{U}_t$ and $\mathcal{U}_{\text{bar},t}$ can be both chosen around $\mathcal{O}(n)$, leading to an overall computational complexity of $\mathcal{O}(n)$. Similar analysis can be carried out for objective equation E2. It is worth noting that all the evaluations of neural networks and localized filter bases can be implemented beforehand, and both optimization objectives are efficiently calculated using basic numerical operations. Table E2 shows the wall clock times for model training with each loss function on all the synthetic data sets, indicating that the computation time for NLL and LS are similar.

## F   ALGORITHM

The weight $w$ and the lower bound $b$ need to be adjusted accordingly during optimization to learn the valid model. For example, non-sensible solutions (negative intensity) appear frequently at the early stage of the training process, thus the lower bound $b$ should be close enough to the minimum of all the evaluations on the grid to effectively steer the intensity functions towards non-negative values, and the weight $w$ can be set as a small value to magnify the penalty influence. When the solutions successfully reach the neighborhood of the ground truth and start to converge (thus no negative intensity would appear with a proper learning rate), the $b$ should be away from the intensity evaluations (*e.g.*, upper-bounded by 0), and $w$ should be large enough to remove the effect of the log-barrier penalty. We provide our training algorithm using stochastic gradient descent in the following.

---

**Algorithm 1** Model learning

---

**Input**: Training set $X$, batch size $M$, epoch number $E$, learning rate $\gamma$, constant $a > 1$ to update $w$ in equation E1 or equation E2.
**Initialization:** model parameter $\theta_0$, first epoch $e = 0$, $s = s_0$. Conditional intensity lower bound $b_0$.
**while** $e < E$ **do**
    Set $b_{temp} = 0$.
    **for** each batch with size $M$ **do**
        1. Compute $\ell(\theta)$, $\{\lambda(t_{c_t}, s_{c_s})\}_{c_t=1,\dots,|\mathcal{U}_{\text{bar},t}|, c_s=1,\dots,|V|}$.
        2. Compute $\mathcal{L}(\theta) = -\ell(\theta) + \frac{1}{w}p(\theta, b_e)$.
        3. Update $\theta_{e+1} \leftarrow \theta_e - \gamma\frac{\partial \mathcal{L}}{\partial \theta_e}$.
        4. Set $b_{temp} = \min\big\{\min\{\{\lambda(t_{c_t}, s_{c_s})\}_{c_t=1,\dots,|\mathcal{U}_{\text{bar},t}|, c_s=1,\dots,|\mathcal{U}_{\text{bar},s}|} - \epsilon, b_{temp}\big\}$, where $\epsilon$ is a small value to guarantee logarithm feasibility.
    **end for**
    $e \leftarrow e + 1, w \leftarrow w \cdot a, b_e = b_{temp}$
**end while**

---

## G   CHOICE OF KERNEL RANK

The kernel rank can be chosen using two approaches. The first approach treats the kernel rank as a model hyperparameter, and its selection is carried out through cross-validation. This process aims to optimize the log-likelihood on validation datasets. Each choice of $(L, R) \in [1, 2] \times [1, 2, 3]$ has been assessed for `GraDK+L3net+NLL` on synthetic and real-world datasets. The order of the $r$-th filter is set to be $r$ (Tables G3 and G4). The log-likelihood values in the synthetic data show only minor differences given different hyperparameter choices once the parameters are chosen such that they sufficiently capture temporal or multi-hop graph influence (typically requiring $R > 1$). In the real-world data, $L = 1$ with $R = 3$ yields the highest log-likelihood on the validation dataset, which are the parameters used in Section 4.2.

The rank of the kernel can also be viewed as a level of model complexity to be learned from the data. Generally, under certain regularity assumptions of the kernel in kernel SVD, the singular values will decay towards 0, resulting in a low-rank kernel approximation. The singular values are captured by

Table G3: Testing log-likelihood for `GraDK` with different kernel ranks.

| Kernel rank | 16-node graph | | | 50-node graph | | | Wildfire | | | Theft | | |
|---|---|---|---|---|---|---|---|---|---|---|---|---|
| | $R=1$ | $R=2$ | $R=3$ | $R=1$ | $R=2$ | $R=3$ | $R=1$ | $R=2$ | $R=3$ | $R=1$ | $R=2$ | $R=3$ |
| $L=1$ | $-3.032_{(<0.001)}$ | $-2.997_{(0.001)}$ | $-2.995_{(0.002)}$ | $-1.085_{(<0.001)}$ | $\mathbf{-1.065}_{(0.002)}$ | $-1.067_{(0.001)}$ | $-3.900_{(0.004)}$ | $-3.654_{(0.009)}$ | $\mathbf{-3.625}_{(0.002)}$ | $-3.405_{(0.001)}$ | $-3.039_{(0.011)}$ | $\mathbf{-2.980}_{(0.003)}$ |
| $L=2$ | $-3.030_{(0.001)}$ | $-2.992_{(0.002)}$ | $\mathbf{-2.987}_{(0.005)}$ | $-1.085_{(0.001)}$ | $-1.067_{(0.002)}$ | $\mathbf{-1.065}_{(0.002)}$ | $-3.977_{(0.005)}$ | $-3.657_{(0.007)}$ | $-3.649_{(0.017)}$ | $-3.396_{(0.008)}$ | $-3.047_{(0.026)}$ | $-3.355_{(0.034)}$ |

*Numbers in parentheses are standard errors for three independent runs. Grey boxes indicate the choice in the original paper.

Table G4: Predictive metrics (Time MAE and Type KLD) for `GraDK` with different kernel ranks.

| Kernel rank | 16-node graph | | | 50-node graph | | | Wildfire | | | Theft | | |
|---|---|---|---|---|---|---|---|---|---|---|---|---|
| | $R=1$ | $R=2$ | $R=3$ | $R=1$ | $R=2$ | $R=3$ | $R=1$ | $R=2$ | $R=3$ | $R=1$ | $R=2$ | $R=3$ |
| $L=1$ | 0.272/0.003 | 0.153/0.001 | **0.028/<0.001** | 1.912/0.004 | **1.023/0.004** | 3.118/0.006 | 0.943/0.107 | 0.358/0.024 | **0.207/0.006** | 2.787/0.097 | 1.052/0.082 | **0.640/0.079** |
| $L=2$ | 0.226/0.002 | 0.145/0.001 | 0.064/0.002 | 3.698/0.006 | 1.024/0.004 | 5.217/0.013 | 0.945/0.109 | 0.314/0.020 | 0.946/0.110 | 2.702/0.091 | 1.001/0.082 | 1.123/0.109 |

*Grey boxes indicate the choice in the original paper.

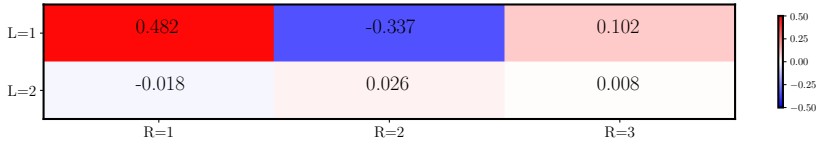

Figure G2: Learned $\{\alpha_{rl}\}$ of `GraDK` with $L=2, R=3$ on wildfire data.

the coefficients $\alpha_{rl}$. The kernel rank is chosen by preserving significant coefficients while discarding higher-order ones; we can treat each $\alpha_{rl}$ as one learnable parameter without the need to choose $D$. The effectiveness of this approach is showcased by the results of the wildfire data. We set $L=2$ and $R=3$ at first and fit the model, and visualize the learned kernel coefficients in Figure G2, which suggests that $L=1$ and $R=3$ would be an optimal choice for the kernel rank. This choice is used in subsequent model fitting.

## H  EXPERIMENTAL DETAILS AND ADDITIONAL RESULTS

In this section, we give details regarding the experiments in Section 4, including a description of the ground truth point process models for synthetic data, latent graph structures for real data, experimental setup, and additional results.

### H.1  DATA DESCRIPTION

Our experiments are implemented using four synthetic and three real-world data sets.

**Synthetic data.**  We provide a detailed description of the ground truth kernels and latent graph structures we used for synthetic data generation:

(i)  3 nodes with non-stationary temporal kernel and 1-hop (positive and negative) graph influence:

$$k(t', t, v', v) = 1.5(0.5 + 0.5\cos(0.2t'))e^{-2(t-t')}\left(0.5 B_1^{(0)}(v', v) + 0.2 B_2^{(1)}(v', v)\right),$$

where $B_1^{(0)} = \mathrm{diag}(0.5, 0.7, 0.5)$, $B_2^{(1)}(2, 1) = -0.2$, and $B_2^{(1)}(2, 3) = 0.4$.

(ii)  16 nodes with non-stationary temporal kernel and 2-hop graph influence:

$$k(t', t, v', v) = 1.5(0.5 + 0.5\cos(0.2t'))e^{-2(t-t')}\left(0.2 B_1^{(0)}(v', v) - 0.3 B_2^{(1)}(v', v) + 0.1 B_3^{(2)}(v', v)\right),$$

such that $(0.2 B_1^{(0)} - 0.3 B_2^{(1)} + 0.1 B_3^{(2)}) = (0.2I - 0.3\tilde{L} + 0.1(2\tilde{L}^2 - I))$. Here $\tilde{L}$ is the scaled and normalized graph Laplacian defined in Section 3.3. This graph influence kernel is visualized in the top row of Figure 1 (first column).

(iii)  50 nodes with exponentially decaying temporal kernel: $g(t', t) = 2e^{-2(t-t')}$. The graph kernel is constructed such that the influence follows a Gaussian density (with 3 modes) along the diagonal of the graph influence kernel with random noise in addition to off-ring influence. The true graph kernel is visualized in the first panel of Figure H6.

(iv)  225 nodes with exponentially decaying temporal kernel: $g(t', t) = 2e^{-2(t-t')}$. The true graph kernel is visualized in the first panel of Figure H7.

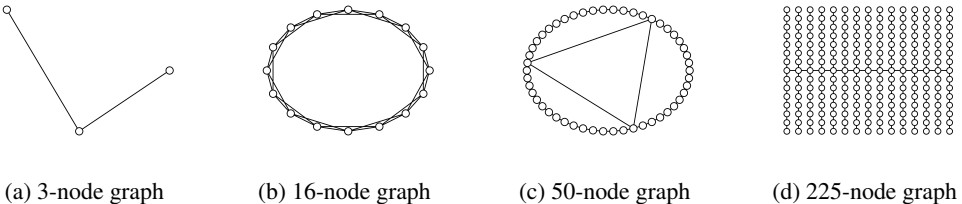

(a) 3-node graph      (b) 16-node graph      (c) 50-node graph      (d) 225-node graph

Figure H3: Latent graph structures for the synthetic data sets. From left to right, the graphs correspond to the 3-node graph, the 16-node ring graph, the 50-node graph, and the 225-noode graph in synthetic data set 1, 2, 3, and 4, respectively.

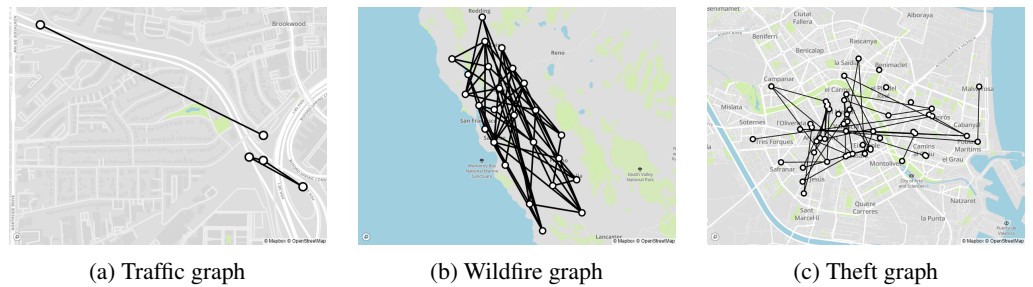

(a) Traffic graph      (b) Wildfire graph      (c) Theft graph

Figure H4: Latent graph structures for the real data sets. From left to right, the graphs correspond to the ones in Atlanta traffic congestion data, the California wildfire data, and the Valencia sustraccion (theft) data.

The latent graph structures for these four synthetic data experiments can be found in Figure H3.

**Real data.** We also apply the model to three real data sets across different real-world settings.

 (i) *Traffic congestion data*. The Georgia Department of Transportation (GDOT) provides traffic volume data for sensors embedded on roads and highways in the state. We have access to such data for 5 sensors at the interchange of interstates 75 and 85 in Midtown Atlanta from September 2018 to March 2019. Traffic volume is measured in 15-minute intervals, and congestion events are detected when the traffic count exceeds the third quartile of the daily traffic volume. The result is 3,830 events which are split into 24-hour trajectories (with an average of 24 events per day). The latent graph connects 5 sensors based on the flow of traffic and proximity.

 (ii) *Wildfire data*. The California Public Utilities Commission (CPUC) maintains a large-scale multi-modal wildfire incident dataset. We extract a total of 2,428 wildfire occurrences in California from 2014 to 2019. The latitude-longitude coordinates of incidents are bounded by the rectangular region [34.51, -123.50] $\times$ [40.73, -118.49]. Note that the majority of the region has no fire in the 5-year horizon due to the fact that fire incidents are likely to cluster in space. Therefore, we apply the K-means algorithm to extract 25 clusters of wildfire incidents. The latent graph is constructed such that each node represents one cluster and is connected to geographically adjacent nodes. The entire dataset is split into one-year sequences with an average length of 436 events.

(iii) *Theft data*. The proprietary crime data collected by the police department in Valencia, Spain records the crime incidents that happened from 2015 to 2019, including incident location, crime category, and distance to various landmarks within the city. We analyze 9,372 sustraccions (smooth thefts) that happened near 52 pubs within the Valencia city area. The graph is constructed from the street network, with each node representing a pub. Two pubs are connected if the distance between them along the street is less than 1 km. Each sustraccion is assigned to the closest pub. We partition the data into quarter-year-long sequences with an average length of 469 events.

The learned latent graph structures overlaid on the real-world geography are displayed in Figure H4 for the Atlanta traffic congestion, California wildfire, and Valencia theft data.

Table H5: Training hyper-parameters for the baselines.

| Model | 3-node synthetic | | 16-node synthetic | | 50-node synthetic | | 225-node synthetic | | Traffic | | Wildfire | | Theft | |
|---|---|---|---|---|---|---|---|---|---|---|---|---|---|---|
| | Learning Rate | Batch Size | Learning Rate | Batch Size | Learning Rate | Batch Size | Learning Rate | Batch Size | Learning Rate | Batch Size | Learning Rate | Batch Size | Learning Rate | Batch Size |
| RMTPP | $10^{-3}$ | 32 | $10^{-3}$ | 32 | $10^{-3}$ | 32 | $10^{-3}$ | 32 | $10^{-3}$ | 32 | $10^{-3}$ | 2 | $10^{-3}$ | 2 |
| FullyNN | $10^{-2}$ | 100 | $10^{-2}$ | 100 | $10^{-2}$ | 100 | $10^{-2}$ | 100 | $10^{-3}$ | 50 | $10^{-3}$ | 20 | $10^{-3}$ | 100 |
| DNSK | $10^{-1}$ | 32 | $10^{-1}$ | 32 | $10^{-1}$ | 32 | $10^{-1}$ | 32 | $10^{-1}$ | 32 | $10^{-1}$ | 2 | $10^{-1}$ | 2 |
| THP-S | $10^{-3}$ | 32 | $10^{-3}$ | 32 | $10^{-3}$ | 32 | $10^{-3}$ | 32 | $10^{-3}$ | 64 | $10^{-3}$ | 2 | $10^{-3}$ | 2 |
| SAHP-G | $10^{-4}$ | 32 | $10^{-4}$ | 32 | $10^{-4}$ | 32 | $10^{-4}$ | 32 | $10^{-4}$ | 32 | $10^{-3}$ | 2 | $10^{-3}$ | 2 |

## H.2 DETAILED EXPERIMENTAL SETUP

We choose our temporal basis functions to be fully connected neural networks with two hidden layers of width 32. Each layer is equipped with the softplus activation function except for the output layer. For each data set, all the models are trained using 80% of the data and tested on the remaining 20%. Our model parameters are estimated through objective functions in Section 3.4 using the Adam optimizer (Kingma & Ba, 2014) with a learning rate of $10^{-2}$ and batch size of 32.

For the baseline of RMTPP, we test the dimension of $\{32, 64, 128\}$ for the hidden embedding in RNN and choose an embedding dimension of 32 in the experiments. For FullyNN, we set the embedding dimension to be 64 and use a fully-connected neural network with two hidden layers of width 64 for the cumulative hazard function, as the default ones in the original paper. The dimension of input embedding is set to 10. For DNSK, we adopt the structure for the marked point processes (for spatio-temporal point processes in the ablation study) and set the rank of temporal basis and mark basis (spatial basis in the ablation study) to one and three. THP-S and SAHP-G use the default parameters from their respective code implementations. The GraDK method uses a temporal basis kernel with rank one and a graph basis kernel with rank three (except a rank two graph basis kernel on the 50-node graph). The temporal kernel basis functions are fully connected neural networks with two hidden layers of size 32. Learning rate and batch size parameters are provided for each baseline and experiment in Table H5.

We note that the original paper of FullyNN (Omi et al., 2019) does not consider the modeling of event marks, and the model can be used for temporal point processes, for it requires the definition of cumulative intensity and the calculation of the derivative, which is inapplicable in event mark space. To compare with this method in the setting of graph point processes, we extend FullyNN to model the mark distribution. Specifically, an additional feedforward fully-connected neural network $\Psi$ is introduced. For each index $i$, it takes the hidden states of history up to $i$-th event $\boldsymbol{h}_i$ as input and outputs its prediction for the likelihood of the next mark $v_{i+1}$ as $\Psi(v_{i+1}|\boldsymbol{h}_i)$. Given a sequence of $n$ events $\{(t_i, v_i)\}_{i=1}^n$, the optimization objective of model log-likelihood for the extended FullyNN is written as:

$$\mathcal{L} = \log L(\{t_i\}) + \log L(\{v_i\})$$
$$= \sum_i \left[\log\left\{\frac{\partial}{\partial\tau}\Phi(\tau = t_{i+1} - t_i|\boldsymbol{h}_i)\right\} - \Phi(t_{i+1} - t_i|\boldsymbol{h}_i)\right] + \sum_i \log\{\Psi(v_{i+1}|\boldsymbol{h}_i)\},$$

where $\Phi$ is the cumulative hazard function (*i.e.*, cumulative intensity function) introduced in FullyNN, parametrized by a feedforward neural network. Such an approach to modeling the distribution of event marks has been commonly adopted in previous studies (Du et al., 2016; Shchur et al., 2019).

## H.3 ADDITIONAL EXPERIMENTAL RESULTS

**Synthetic data.** Figure H5 presents the kernel and intensity recovery results for the synthetic data set 1 generated by a non-stationary temporal kernel and a 3-node-graph kernel with inhibitive influence. The recovery outcomes demonstrate the efficacy of our proposed model in characterizing the temporal and graph-based dependencies among events, as evident from the first and second rows of the figure. Moreover, the results emphasize the capability of our model, GraDK, to effectively capture the inhibitive effects of past events. In contrast, the event dependencies represented by the normalized positive attention weights in SAHP solely capture the triggering intensity of past events without accounting for inhibitive influences. Similarly, the first row of Figure H6 displays the true

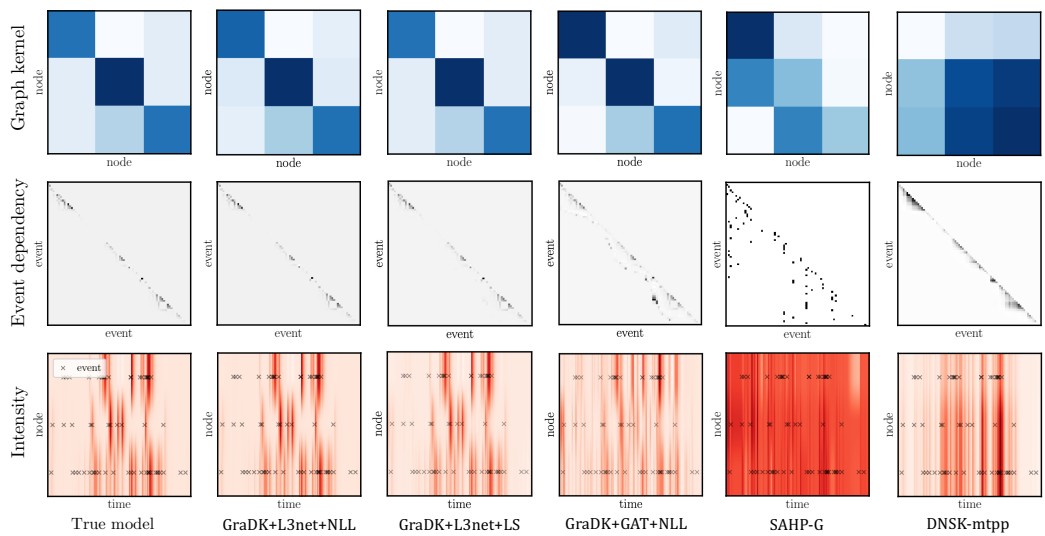

Figure H5: Graph kernel, inter-event dependence, and conditional intensity recovery for the 3-node-graph synthetic data set with negative (inhibitive) graph influence. The first column reflects the ground truth, while the subsequent columns reflect the results obtained by `GraDK` (our method), `SAHP-G`, and `DNSK`, respectively.

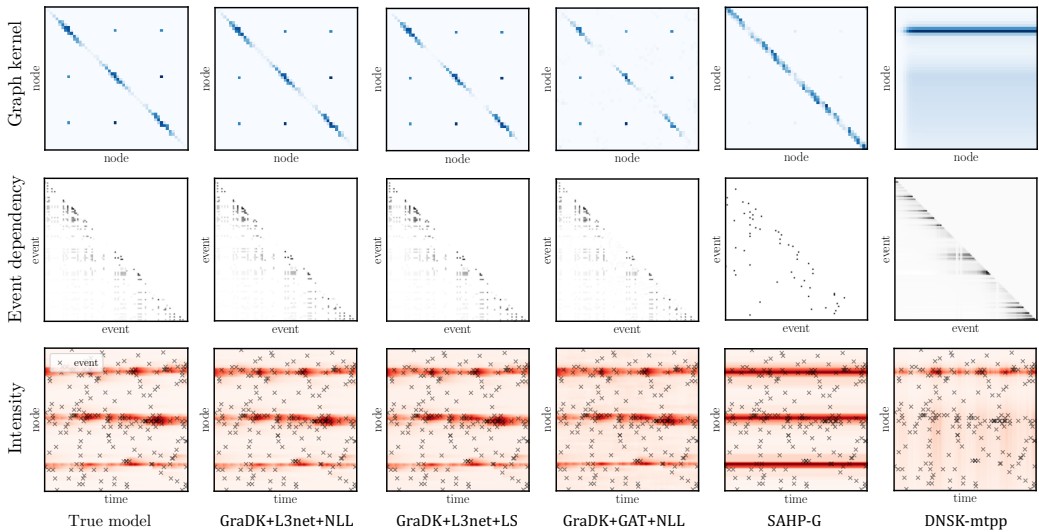

Figure H6: Graph kernel, inter-event dependence, and conditional intensity recovery for the 50-node synthetic data set.

graph influence kernel in the 50-node synthetic data set and the learned graph kernels by `GraDK`, `SAHP-G`, and `DNSK`. While `SAHP-G` exaggerates the self-exciting influence of graph nodes and `DNSK` only learns some semblance of the graph kernel behavior, our method accurately recovers both self- and inter-node influence patterns, resulting in a faithful model of the true graph kernel. The conditional intensity via each method for one testing trajectory is displayed in the third row of Figure H6, which demonstrates the capability of our model to capture the temporal dynamics of events.

**Real data.** We visualize the learned graph kernels by `GraDK`, `SAHP`, and `DNSK` on traffic and wildfire data in Figure H8. Our approach is able to learn intense graph signals with complex patterns by taking advantage of the graph structure and GNNs. While the attention structures adopted in `SAHP` contribute to improved model prediction performance for future events, this approach suffers

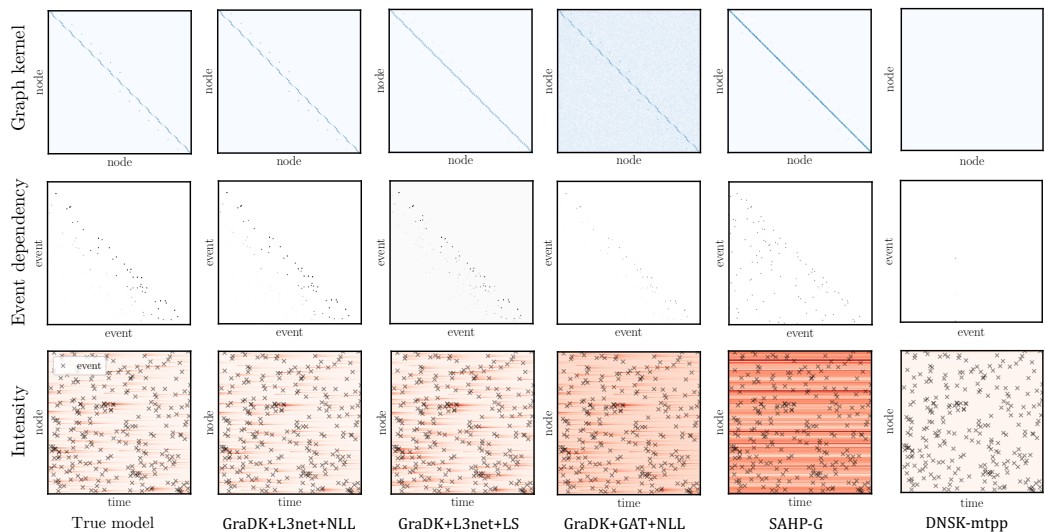

Figure H7: Graph kernel, inter-event dependence, and conditional intensity recovery for the 225-node synthetic data set.

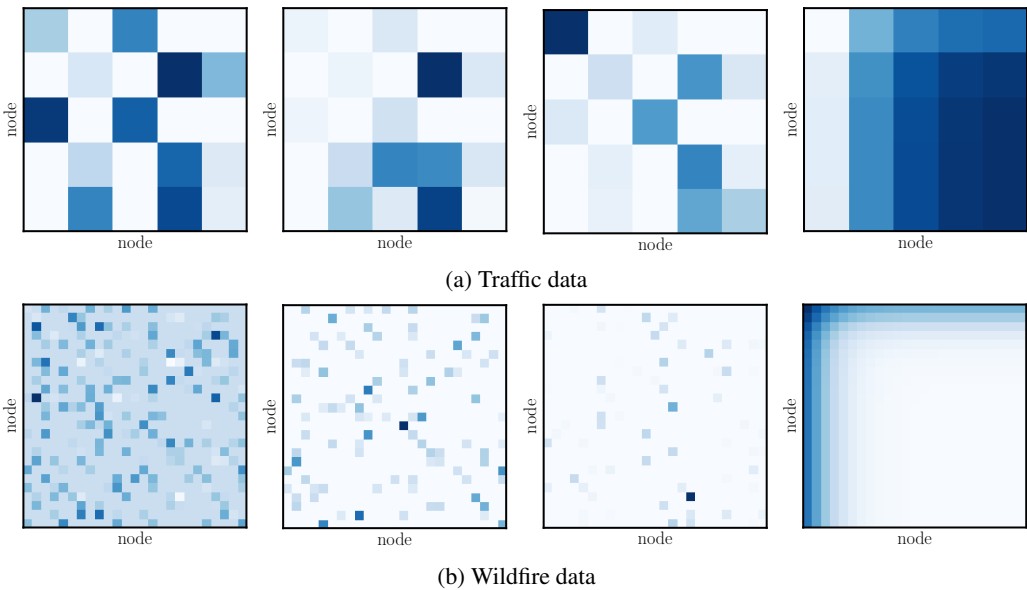

Figure H8: Learned graph kernels for traffic and wildfire data set. The columns present the recovered kernels on each data set of `GraDK+L3net`, `GraDK+GAT`, `SAHP-G`, and `DNSK`, respectively.

from the limited model expressiveness and interpretability when attempting to recover the underlying mechanism of the event generation process, indicated by the weak and noisy graph signals. `DNSK` fails to uncover the intricate patterns existing in graph kernels and only provides restrictive kernels for event modeling without considering the latent graph structures among data.

Our model not only achieves exceptional interpretability but also holds practical significance in the context of real-world point process data modeling. This can be demonstrated through experimental results conducted on traffic data. The five traffic sensors from which we collect data can be categorized into two groups, northbound sites and southbound sites, according to the directions of the highway they are monitoring. Figure H9(a) visualizes the structure of the traffic network with sensors (graph nodes) labeled and arrows indicating the highway directions. We then investigate the conditional intensity learned by `GraDK` on each traffic sensor. Figure H9(b)(c) show the conditional intensity

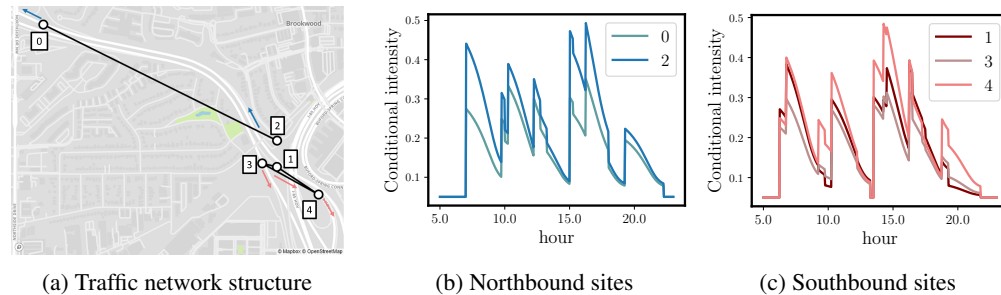

|                              |                          |                          |
| :--------------------------: | :----------------------: | :----------------------: |
| (a) Traffic network structure | (b) Northbound sites    | (c) Southbound sites     |

Figure H9: (a) Traffic network structure. Each traffic sensor is labeled with a number. The blue and pink arrows indicate the monitored traffic directions of sensors on northbound and southbound highways, respectively. (b)(c) Conditional intensity of five sensors in a single day, which are categorized into two groups according to the monitored traffic directions of the sensors.

functions of five sensors during one single day (*i.e.*, computed given one sequence from the testing set) with two subgroups of northbound and southbound sites. Note that similar temporal patterns can be found within the same subgroup, which can be attributed to the fact that the sensors in the same group are in the same direction and share the same traffic flow. Also, all the intensity functions show a temporal pattern in which they reach pinnacles during the morning (around 8:00) and evening (around 17:00) rush hours, with a higher possibility for traffic congestion at southbound sensors in the afternoon.

