# OpenReview forum: "Deep graph kernel point processes"
_ICLR.cc/2024/Conference — Submitted to ICLR 2024_

### Official Review · Reviewer_XHVx · 2023-10-30

**Soundness:** 3 good
**Presentation:** 3 good
**Contribution:** 2 fair
**Rating:** 5
**Confidence:** 3

**Summary:**

This paper introduces GNN into the spatio-temporal point process framework to enhance the expression ability of point processes on graph data. The idea of introducing graph neural networks to model the kernel of point processes is novel. The experimental results sufficiently support the effectiveness.

**Strengths:**

1. The paper is well-written and easy to follow.
2. The idea of modeling kernels in point processes by GNNs is novel and meaningful to the development of the community. The introduction of localized graph filters improves the scalability of our model when applied to large graphs.
3. The experiments are sufficient and support the main idea of this paper.

**Weaknesses:**

The main concern is the contribution of this article. Many previous works have tried to study point processes in the context of GNN. (For example, see [1]) From this perspective, this article is not innovative enough. Among them, the main idea of equation (2) comes from [2]. This article focuses on expressing the h_d function using localized graph filters. Although effective, the contribution appears to be limited.


[1] Pan Z, Wang Z, Zhe S. Graph-informed Neural Point Process With Monotonic Nets[J]. 2022.
[2] Dong Z, Cheng X, Xie Y. Spatio-temporal point processes with deep non-stationary kernels[J]. arXiv preprint arXiv:2211.11179, 2022.

**Questions:**

1. Based on comparison with existing work, would you mind further elaborating on the contribution of this article?

---

> ### Author Response · Authors · 2023-11-20
> **Detailed comparison with literature**
>
> We thank the reviewer for the positive comments and confirmation of our experiments.
>
> Regarding weakness, we have now provided a comprehensive comparison with related works in the literature, in Table A.1, and have explained the contribution/novelty of our paper in detail in Appendix A.
>
> We want to point out that the extension to the graph kernel is non-trivial, and a lot of modeling and experiment validation development needs to be done. We have considered various types of graph kernels based on various GNNs and also consider the case when the graph topology is known, and when the graph topology is unknown, then we use the GAT. We provided extensive numerical experiments and compared them with multiple baselines and SOTA. In particular, since point process data over graph is so widely available, and there is still a lack of principled graph kernel design, we believe this work will be a valuable addition to the literature.

---

### Official Review · Reviewer_nvzU · 2023-11-01

**Soundness:** 2 fair
**Presentation:** 3 good
**Contribution:** 2 fair
**Rating:** 5
**Confidence:** 3

**Summary:**

This paper proposes using graph neural networks to define the inter-mark relationship in triggering kernels of marked Hawkes processes. The proposed model represents the triggering kernels by localized graph filter bases, which permit flexible modeling of inter-event-category dependence, including non-stationary, multi-hop exciting, and inhibiting effects. The validity of the model was confirmed by the experiments on synthetic and three real-world data.

**Strengths:**

- The paper is well-written and easy to follow. The relationship with the related works is clearly presented.
- Experimental results support that the proposed model provides good predictive performance.
- The training algorithm scales linearly with the data size.
- The validity of the proposed model was evaluated on several real-world data.

**Weaknesses:**

- The good accuracy achieved by the proposed model is practically important, but the technical contribution of the model seems to be somewhat marginal because it is a reasonable but mediocre idea to use a GNN to model the inter-mark relationship in inference kernels, and any technical difficulty is not seen in the training algorithm.
- Any limitations of the proposed model are not discussed. For example, it seems that the intensity function of the proposed model could be negative even if the log-barrier method is adopted, while the conventional methods (e.g., RMTTP) are designed not to worry about it.
- (Minor comment) There are typos in Reference (e.g., rules of upper/lowercase is not consistent).

**Questions:**

- To the best of my knowledge, (Omi et al., 2019) doesn’t consider marks of each event in the model. How was FullyNN implemented in the experiment as a marked point process model?

---

> ### Author Response · Authors · 2023-11-20
> **Technical novelty**
>
> We thank the reviewer for the positive comments. We now have clarified in Appendix A that incorporating GNN in the graph-based influence kernel is crucial and non-trivial, requiring a carefully designed model. Our paper is the first one to incorporate GNN in the influence kernel without any parametric constraints.

---

> ### Author Response · Authors · 2023-11-20
> **Discussing limitation of proposed model**
>
> We now have discussed the potential limitation in the Conclusion of the paper. As we have explained to the above reviewer, current kernel representation still assumes that the influence over events is "additive," which is common in the kernel-based type of methods for point processes. To be more general, one can potentially adopt a non-additive model over events that can be more complex, and we leave it for future work.

---

> ### Author Response · Authors · 2023-11-20
> **Ensuring non-negativity of intensity function of the proposed model**
>
> In our paper, in our log-barrier function equation in the appendix (E.1) and (E.2), we used a small constant b<0 inside the log-barrier, which means that we can allow the intensity to have small non-negativity. However, we do this to get an easy initialization: we can start with a reasonable guess that allows for a small amount of negativity and tune b to diminish to 0 as the training process continues, and in the end, we observe in the experiment in the end the learned intensity function induced by the log-barrier is always non-negative.

---

> ### Author Response · Authors · 2023-11-20
> **Typos in Reference (e.g., rules of upper/lowercase is not consistent)**
>
> Thank you very much. We have fixed these typos in the reference.

---

> ### Author Response · Authors · 2023-11-20
> **Comparison with (Omi et al., 2019)**
>
> Indeed, the reference (Omi et al. 2019) did not consider marks, and their model can only be used for temporal point process. In order to compare with this method for the graph setting, we further extend their method to model the mark distribution on top of FullyNN. We have now provided complete details of the extended FullyNN in Appendix H.2.

---

> > ### Comment · Reviewer_nvzU · 2023-11-20
> > **Response to author comments**
> >
> > Thank you for the authors’ clarification. I stand by the point that the technical contribution of the paper is valuable but not impactful, and keep my score.
> >
> > Comments to the discussion about non-negativity of intensity: I understand that no negative intensity values were observed in the experiments, but it necessarily ensures that the learned intensity function model does not have any negative values on a new event data. For example, if unexpectedly frequent events are emitted from a mark who has a negative impact on a new test data, then the intensity function could be negative. I believe the practical values of the proposed model, but some caution should be exercised when using it.

---

> > > ### Author Response · Authors · 2023-11-23
> > > **Reply to Reviewer nvzU's comment**
> > >
> > > We thank the reviewer for the further comments. We would like to make a kind reminder that we have uploaded the updated version of our paper, where we have addressed the limitation, and provided a discussion to the question regarding "ensuring positivity by log-barrier" in the last section of the paper.
> > >
> > > We opt for the log-barrier in model estimation due to its computational efficiency and its ability to preserve the model interpretability for the additive event influence. In practice, ensuring the non-negativity of the intensity using log-barrier is achievable when the occurrences of new events follow the same probability distribution as the observed ones, aligning with the fundamental principle in machine learning. However, we agree that potential distributional shifts may give rise to frequent future events with negative impacts, which could result in a negative intensity. Such a possibility is worth considering, and we will leave it as a topic for future research.
> > >
> > > We hope our response appropriately addresses your concerns. We greatly appreciate your time and feedback on our work.

---

### Official Review · Reviewer_o26h · 2023-11-02

**Soundness:** 3 good
**Presentation:** 2 fair
**Contribution:** 3 good
**Rating:** 6
**Confidence:** 5

**Summary:**

The paper presents a novel point process model for discrete event data over graphs. It extends the influence kernel-based formulation by Hawkes to include Graph Neural Networks (GNN), aiming to capture the influence of historical events on future events' occurrence. The model combines statistical and deep learning approaches to enhance model estimation, learning efficiency, and predictive performance. The paper also demonstrates the superiority of the proposed model over existing methods through comprehensive experiments on synthetic and real-world data.

**Strengths:**

1.	The paper introduces an innovative approach by integrating GNN with the classic influence kernel-based formulation, offering a new perspective in point process modeling over graphs.
2.	The experimental setup and the methodologies used are sound and well-executed.

**Weaknesses:**

1.	The proposed approach appears to be a combination of existing methods, which raises questions about its novelty. To provide a valuable contribution to the field, it is crucial to address the limitations and shortcomings of existing approaches, such as kernel-based, deep neural network-based, and graph-based models. Additionally, the authors should clearly articulate the advantages of this new method compared to GNN-based models [1,2,3,4].

- [1] Learning neural point processes with latent graphs，WWW2021
- [2] Graph neural point process for temporal interaction prediction，TKDE2022
- [3] Spatio-temporal point processes with deep non-stationary kernels，ICLR2022
- [4] Gaussian process with graph convolutional kernel for relational learning，KDD2021

2.	Graph construction, including the definition of nodes and edges, is pivotal in GNN-based methods. The paper needs to provide a detailed explanation of how edges between nodes are established. Are the edge construction methods applicable universally across different scenarios, or do they require case-specific adaptations? While the process is well-defined for synthetic datasets, it is less clear in real-world data scenarios. The assertion that edges represent potential interactions should be elaborated upon to ensure clarity.
3.	The authors use THP-S and SHAP-G as baselines. However, these baselines employ powerful transformers and graph attention mechanisms. It would be expected that they offer significant flexibility. This work should provide a robust justification for why the proposed model outperforms these baselines. Is there a theoretical basis for the superior performance?
4.	Notably, the synthetic data is primarily generated using kernel methods. Hence, I think the proposed model should fit this data well. However, the observed improvement in likelihood compared to the baseline is very minimal and possibly not statistically significant. The authors should offer a convincing explanation for this limited improvement.
5.	This concern follows question 4. The discrepancy between the log-likelihood, which is similar to the baseline, and the substantially improved time prediction MAE is perplexing. I am suspicious about the accuracy of the MAE calculation. I recommend the authors clarify how this probabilistic prediction model is employed for predictions – whether it is based on random sampling or mean predictions.
6.	In the evaluation using real-world data, Table 3 indicates that the advantage of the graph-based approach diminishes as the number of nodes increases. This observation runs counter to my expectation that graph modeling should excel in complex graph structures. The authors should provide a detailed analysis and potential explanations for this outcome.
7.	Lack of crucial baseline comparisons. A robust baseline comparison is essential for a comprehensive evaluation of the proposed method. The absence of such comparisons is a notable limitation in the paper. For instance, even without incorporating a graph, models like THP [5], and DAPP[6] possess the inherent capability to capture inter-event interactions. Additionally, there are intensity-free methods [7] available in the literature. Hence, it is necessary to compare the proposed model against state-of-the-art point process models [1,2,3,4,5,6,7].
- [5] Zuo, Simiao, et al. "Transformer hawkes process." International conference on machine learning. PMLR, 2020.
- [6] Zhu, Shixiang, et al. "Deep fourier kernel for self-attentive point processes." International Conference on Artificial Intelligence and Statistics. PMLR, 2021.
- [7] Shchur, Oleksandr, Marin Biloš, and Stephan Günnemann. "Intensity-Free Learning of Temporal Point Processes." International Conference on Learning Representations. 2019.
8.	Similarly, in real-world data evaluation, there is an absence of comparisons with state-of-the-art STPP models and their mentioned baselines. The authors should consider referring to recent literature [8,9,10] for these comparisons to provide a more comprehensive evaluation.
- [8] Zhou, Zihao, et al. "Neural point process for learning spatiotemporal event dynamics." Learning for Dynamics and Control Conference. PMLR, 2022.
- [9] Chen, Ricky TQ, Brandon Amos, and Maximilian Nickel. "Neural Spatio-Temporal Point Processes." International Conference on Learning Representations. 2020.
- [10] Yuan Yuan, et al. “Spatio-temporal Diffusion Point Processes.” In Proceedings of the 29th ACM SIGKDD Conference on Knowledge Discovery and Data Mining (KDD '23).

## After rebuttal
I think the authors have addressed most of my concerns, as the added Appendix A provides a detailed comparison with related works.

**Questions:**

Please see my listed weakness above.

---

> ### Author Response · Authors · 2023-11-12
> **Novelty**
>
> We thank the reviewer for the comment. We have clearly summarized the novelty of the method, which is not simply an addition of existing methods in fact, there are non-trivial challenges to overcome and carefully design the model considering the graph structure of the data in a principled manner while maintaining the statistical/stochastical law of the point process, which is achieved through the kernel representation. Moreover, as also mentioned by the other reviewer, the core novelty is "The idea of introducing graph neural networks to model the kernel of point processes is novel." "The idea of modeling kernels in point processes by GNNs is novel and meaningful to the development of the community." We will also provide answers to the remaining questions.

---

> ### Author Response · Authors · 2023-11-23
> **Response to Reviewer o26h**
>
> > The proposed approach appears to be a combination of existing methods, which raises questions about its novelty.
>
> We have now provided a comprehensive comparison with related works in the literature in Table A.1, and have explained the contribution/novelty of our paper in detail in Appendix A.
>
> > The paper needs to provide a detailed explanation of how edges between nodes are established.
>
> We would clarify that for real-world datasets, we adopt GAT in our framework to infer the underlying graph structure and node interactions based on the learned attention weights from data, since the underlying graph structure is unobserved in real-world datasets and we have no prior knowledge about the graphs. The latent graph structure is constructed upfront in synthetic datasets.
>
> > Is there a theoretical basis for the superior performance against the THP-S and SHAP-G baselines? These baselines employ powerful transformers and graph attention mechanisms.
>
> We first would point out that the model performance of neural point processes is affected by various factors (*e.g.*, deep neural network architectures, model formulation of the point process, specific problems), and so far, a general theoretical basis can hardly be established.
>
> Meanwhile, we would clarify there is no GNN architecture incorporated in \texttt{THP-S} and \texttt{SHAP-G}, and they model the conditional intensity function instead of the influence kernel, which may not be able to capture the repeated influence patterns effectively like our method.
>
> > The observed improvement in likelihood compared to the baseline is very minimal and possibly not statistically significant.
>
> We clarify that we not only report the average metrics from multiple independent runs but also the standard errors of those metrics. We believe that the improvement is significant both from statistical and practical perspectives.
>
> > Clarify how this probabilistic prediction model is employed for predictions – whether it is based on random sampling or mean predictions.
>
> The predictability of the proposed model is evaluated through sampling. As we clarify in the paper, we use the learned model on the training set to generate new event sequences over the entire time horizon, and compare the generated sequences with the testing sequences, in terms of the predicted event frequency and the predicted distribution of event types. The time MAE is calculated as the mean absolute error between the temporal frequency of generated events and testing events.
>
> > Table 3 indicates that the advantage of the graph-based approach diminishes as the number of nodes increases, which runs counter to my expectation.
>
> We would point out that the comparison of the absolute difference of performance metrics across different point process datasets has little practical meaning, for they have diverse problem settings and data features. More importantly, our proposed method enjoys superior performance advantages against baselines consistently across different datasets.
>
> > Lack of crucial baseline comparisons.
>
> We would highlight that we have already compared with several state-of-the-arts that the reviewer listed, and have highlighted the results in the paper's experiments (Section 4).
>
> > Similarly, in real-world data evaluation, there is an absence of comparisons with state-of-the-art STPP models and their mentioned baselines. The authors should consider referring to recent literature for these comparisons to provide a more comprehensive evaluation.
>
> We would highlight that we have compared with state-of-the-art STPP models in the ablation study (Section 4.2.1), which has clearly demonstrated the superior performance of our method against modeling graph point process using models constructed in the Euclidean space.

---

### Official Review · Reviewer_E4mM · 2023-11-07

**Soundness:** 4 excellent
**Presentation:** 4 excellent
**Contribution:** 4 excellent
**Rating:** 8
**Confidence:** 4

**Summary:**

This paper proposes a flexible deep graph kernel point process model for marked point process data, where marks are treated as "nodes" in a graph. The interactions between marks are modeled through Graph Neural Networks (GNN), which are very flexible to capture various types of interactions over a graph. The kernel representation of the marked point process is then constructed through a convolution of kernels over the time domain and the graph. The effectiveness of the proposed method is demonstrated through extensive simulation studies and some benchmark data points, which outperforms many existing methods for marked point process data.

**Strengths:**

I found the idea of treating marks as graph nodes to be very interesting, which opens the door for many future research directions. The proposed kernel representation of the marked point process is straightforward but powerful, suggesting a wide range of possible applications. The paper is very well written and the presentation is clear. The numerical experiments are impressive and convincing.

**Weaknesses:**

It will be good to add some background information on the graph-based kernels so that the paper is more self-contained. It will also be helpful if the authors can comment on what types of network interactions can be modeled and what the limitations of such representation are. This way, the readers will have a better idea of the advantages and limitations of the proposed model.

**Questions:**

1. Some relevant recent work on Hawke's process on networks is missing. For example, [1]. In [1], the graph structure (i.e., the adjacency matrix) is observed, can you comment on how one can use this information in the proposed model? [1] also handles the heterogeneity of the nodes in the graph, can similar things be done using the proposed model?

2. As I mentioned before, it will be good to add some background information on the graph-based kernels so that the paper is more self-contained. It will also be helpful if the authors can comment on what types of network interactions can be modeled and what the limitations of such representation are. This way, the readers will have a better idea of the advantages and limitations of the proposed model.

3. On page 4, the basis functions of $g_d(\cdot,\cdot)$ typically need to be orthogonal. Can you comment on how to ensure the orthogonality of basis functions using the fully connected neural network? If orthogonality is not imposed, how do you ensure the uniqueness of the decomposition?

Reference:

[1]. Fang, G., Xu, G., Xu, H., Zhu, X., & Guan, Y. (2023). Group network Hawkes process. Journal of the American Statistical Association, (just-accepted), 1-78.

---

> ### Author Response · Authors · 2023-11-20
> **Add some background information and kernel representation**
>
> We thank the reviewer for the positive comment and acknowledge the value of our proposed model. We have added additional background information on the graph-based kernels to make the paper more self-contained.
>
> Regarding "what types of network interactions can be modeled and what the limitations of such representation are." Currently, we assume the kernel is not "rank-one," with more than one pair of basis functions (over time and graph), the temporal kernel can allow non-stationarity over time, and the graph basis is completely general to enable efficient computational; we assume each basis function is decoupled over "time" and "graph" which is also intuitive and commonly used in the literature; compared with prior models, which usually is "rank-one" our model can be much more expressive while striking similar computational complexity.
>
> One potential limitation is that the current kernel representation still assumes that the influence over events is "additive," which is common in the kernel-based type of methods for point processes. To be more general, one can potentially adopt a non-additive model over events that can be more complex, and we leave it for future work. We have now discussed the limitation in the Conclusion section.

---

> ### Author Response · Authors · 2023-11-20
> **Graph structure**
>
> We want to emphasize that our framework can directly incorporate known graph structure; in particular, the observed graph structure information can be leveraged in the construction of the localized graph filter $B_r(v', v)$ in the influence kernel based on the specific formulation of the GNN structures discussed in Appendix CB (\textit{e.g., Chebnet, GAT, L3net, GPS Graph Transformer}). Indeed, we have demonstrated this using synthetic data experiments in Section 4.1.2; the localized graph filters are created based on the observed graph structures.
>
> We currently do not directly handle
> ``heterogeneity of nodes in graph,'' we admire this approach in [1] and believe our model can potentially be extended in the future to account for this, for instance, by introducing latent structures to the nodal feature.

---

> ### Author Response · Authors · 2023-11-20
> **Basis function orthogonality**
>
> Although the design of the kernel is motivated by Mercer's theorem explained in Sec. 3.2, and the basis function is represented using neural networks. In practice, we do not need orthogonality for the basis function. The reason can be explained potentially by the fact that as long as the basis function spans the space, it will be a good representation of the kernel; the training of the neural networks is done using stochastic gradient descent, which rarely leads to any nearly linearly dependent basis functions. In our experiment, we have chosen the rank $r$ to be large enough such that the empirical performance is good and validated in Appendix G.

---

### Author Response · Authors · 2023-11-12
**Confused by some questions of Reviewer o26h**

We are confused by several questions raised by reviewer o26h. For example, it was asked that we did not compare with a list of works; however, we indeed have already compared with several of the paper mentioned and stated clearly in the experiment section in the paper. For example, the following papers

[3] Spatio-temporal point processes with deep non-stationary kernels, ICLR2022

We have compared this method under different real-data settings, and the results are in Table 2; more comparisons with this method are shown in Tables 1 and 3.

[5] Zuo, Simiao, et al. "Transformer hawkes process." International conference on machine learning. PMLR, 2020.

In Tables 1 and 3, we compared them with [5] and showed much better performance.

As the rest of the reviewers (3 out of 4 reviewers) stated that the paper is "clear" and "well-written," "The relationship with the related works is clearly presented," we do not believe that this is because we did not explain this clearly in the paper.

So we are very confused and not sure if the comments are based on reading of our paper.

---

### Author Response · Authors · 2023-11-20
**Paper update**

We thank all the reviewers for their thoughtful reviews and valuable feedback. We have revised our paper to address questions from the reviewers. The major changes are highlighted in blue and summarized below: (R1=E4mM, R2=o26h, R3=nvzU, R4=XHVx)

- Following all the reviewers' suggestions, we have revised the Related work (Section 1.1) and added one section of Appendix A to further elaborate on the contributions of our paper, where we provide a detailed comparison between our method and existing approaches that study the point processes in the context of graphs, neural networks, and graph neural networks. The references mentioned by R1, R2, and R4 have been reviewed in the revised paper.

- Following R1's suggestions, we have included additional background information on the graph-based kernel in Section 2, together with the comprehensive background of the kernel and localized graph filters in Appendix C.

- Following R1 and R3's suggestions, we have discussed the potential model limitations in Section 5 (Conclusion).

- Following R3's question, we have added details of the extension of FullyNN to the setting of graph point processes in Appendix H.2.

Other minor changes have also been made to fix the notation issues and improve the writing.

---

### Author Response · Authors · 2023-11-23
**Dear AC and reviewers**

We hope that our responses have appropriately addressed your concerns. Please let us know if you have any additional questions or comments. We would be more than happy to follow up with additional details. Thank you for dedicating your time to reviewing our efforts.

---

### Meta-Review · Area_Chair_q8Qx · 2023-12-06

**Metareview:**

This paper introduces a method called GraDK (deep graph kernel), which utilizes a graph neural network to learn influence kernels for point processes. The construction makes use of kernel singular value decomposition (SVD) and the rank choice is carried out via cross-validation. Using a flexible and powerful graph neural network is an interesting idea to tackle problems relating to the point process. However, from the reviewers' discussion, one limitation is the technical contributions of existing methodologies, which could be further explained and clarified.

Throughout the discussion period, there has been a lively exchange of ideas, leading to an improvement in the manuscript, which includes discussions on the proposed method's relationship with various existing techniques as shown in Appendix A. While direct comparisons or extensive discussions for methods in Appendix A, which haven't been conducted yet, can extensively strengthen the manuscript, this promising submission could benefit from additional refinement, considering the valuable insights gained from these discussions.

**Justification For Why Not Higher Score:**

Novelty is limited

**Justification For Why Not Lower Score:**

N/A

---

### Decision · Program_Chairs · 2024-01-16

Reject